# RePro: Training Language Models to Faithfully Recycle the Web for Pretraining

## Abstract

High-quality data is a cornerstone of large language model (LLM) pretraining, yet its growth has not kept pace with the needs of frontier models. In this paper, we introduce REPRO, a novel web recycling method that trains a relatively small LM with reinforcement learning to generate effective and faithful rephrasings of pretraining data. Specifically, we design one *quality* reward and three *faithfulness* rewards, optimizing the LM rephraser to convert organic data into high-quality rephrasings while maintaining its core semantics and structure. In our experiment, we train a 4B rephraser to recycle 72B tokens sampled from DCLM-RefinedWeb. Pretraining results on 400M, 1.4B, and 2.8B models demonstrate that REPRO delivers 4.7%-14.0% relative accuracy gains over organic-only baseline on 22 downstream tasks. REPRO also outperforms ReWire, the state-of-the-art web recycling method that prompts a 70B rephraser, as well as the organic baseline with a 4× larger data pool. Experiments with different amounts of recycled data highlight that REPRO improves organic data efficiency by 2-3×. Individual and distributional analyses validate that REPRO preserves more critical information and faithfully reflects the characteristics of organic data compared to prompting-based methods. Together, these results show that REPRO provides an efficient and controllable path to effectively harness organic pretraining data. Our anonymized code is available at https://anonymous.4open.science/r/RePro. We will open-source our rephraser and recycled data.

## 1 Introduction

The continued scaling of large language models (LLMs) is threatened by the diminishing supply of high-quality pretraining data (Villalobos et al., 2024; Maini et al., 2025). While the web provides vast amounts of content, referred to as organic data, standard data pipelines often filter out the majority of it as "low-quality" to ensure pretraining quality (Weber et al., 2024; Li et al., 2024). The shortage of high-quality data thus leads to a looming "data wall" that impedes further progress in LLM pretraining (Nguyen et al., 2025).

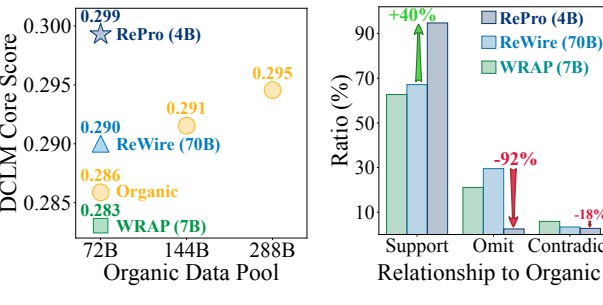

(a) Downstream performance     (b) Key point recall

Figure 1: (a) Pretraining performance of 1.4B model and (b) ratio of key points in organic data that are supported / omitted / contradicted in the recycled version across three web recycling methods (rephraser size in parentheses).

To address the scarcity of high-quality data, one promising path is to "*recycle*" the low-quality web data by rephrasing, thereby increasing the amount of usable pretraining data (Nguyen et al., 2025). Previous approaches by prompting LLMs (e.g., Llama-3.3-70B-Instruct (Dubey et al., 2024)) have shown promising results, matching or even surpassing the performance of doubling organic data (Maini et al., 2024; Nguyen et al., 2025). However, these methods face two major limitations. First, the computational cost of rephrasing with large models is prohibitively high; second, prompts alone may not faithfully preserve the semantics and structure of organic data (Bi et al., 2025), which is critical for the reliability

and richness of the pretraining corpus. These challenges underscore the need for a cost-efficient and faithful web recycling method for LLM pretraining.

In this paper, we introduce REPRO, an efficient and faithful web recycling method to effectively harness organic pretraining data. Our approach leverages reinforcement learning (RL) to train a relatively small rephraser using two categories of rewards: *quality* and *faithfulness*. Specifically, we choose DataMan score (Peng et al., 2025), the state-of-the-art quality assessment metric, as the quality reward to incentivize high-quality rephrasings. For faithfulness rewards, we employ BERTScore (Zhang et al., 2020; Zhu & Hauff, 2021), structure preservation, and length alignment to maintain the semantic meaning, structural diversity, and length distribution of organic data. Together, these reward functions guide our rephraser to produce high-quality pretraining data while faithfully preserving the characteristics of organic data.

We train our rephraser from Qwen3-4B (Yang et al., 2025), and employ it to recycle a 72B organic data pool sampled from DCLM-RefinedWeb (Li et al., 2024; Nguyen et al., 2025). We pretrain 400M, 1.4B, and 2.8B models on the high-quality portion of organic and recycled data. Evaluation results on 22 downstream tasks, covering reasoning, understanding, and knowledge, demonstrate that REPRO achieves +4.7% to +14.0% average accuracy (DCLM Core score) gains over the organic baseline. As shown in Figure 1a, REPRO also outperforms (1) the state-of-the-art web recycling method ReWire, which prompts Llama-3.3-70B-Instruct with chain-of-thought reasoning, and (2) organic baseline when enlarging the data pool by 4×. We also increase the amount of recycled data in pretraining and find that REPRO can boost organic data efficiency by 2-3×—the proportion of high-quality data in the final pretraining set compared to high-quality organic data.

Further ablation studies show that our reward functions effectively optimize the rephraser to generate high-quality data while maintaining the semantic meaning and structural diversity of organic data. As a representative study, we measure how many key points in one organic data are supported, omitted, or contradicted in its recycled version using the method from Qi et al. (2024). As shown in Figure 1b, REPRO achieves the highest number of 95% supported key points while significantly reducing the ratio of omitted key points by up to 92% compared to prompting-based methods WRAP (Maini et al., 2024) and ReWire. In-depth analysis of rephrasing operations shows that RE-PRO flexibly applies diverse operations such as paraphrasing, removing, and clarifying to enhance data quality. These results highlight that REPRO recycles web data in a faithful manner, effectively alleviating the data scarcity issue in LLM pretraining.

We summarize the main contributions of our work as follows:

1. We propose REPRO, a novel web recycling method for better organic data efficiency by training a relatively small LM with RL to perform effective and faithful rephrasing.

2. We design one quality and three faithfulness rewards to optimize our rephraser to generate high-quality data while maintaining the semantics, structure, and length of organic data.

3. REPRO outperforms the state-of-the-art recycling method (ReWire) despite using a 17× smaller rephraser and boosts the organic data efficiency by 2-3×. Analyses confirm that REPRO faithfully preserves the essential semantics and structure of organic data.

## 2  RELATED WORK

The scaling of large language models (LLMs) has expanded along three primary dimensions: model size, compute budget, and pretraining data volume (Kaplan et al., 2020; Hoffmann et al., 2022). In the early stages of LLM development, the main bottleneck was compute, which has since been largely alleviated by advances in hardware efficiency, novel model architectures (Shazeer et al., 2017; Liu et al., 2024), and the selective use of high-quality data (Engstrom et al., 2024; Wettig et al., 2024; Yu et al., 2024). Recent forecasts indicate that another critical constraint lies in the quantity of pretraining data, as the supply of organic (human-generated) text available on the internet is expected to be rapidly exhausted (Villalobos et al., 2024). Meanwhile, standard data curation pipelines often discard a large portion of the collected web data to ensure the quality of pretraining (Weber et al., 2024; Li et al., 2024). These practices highlight the scarcity of high-quality data, which has become a new bottleneck for further scaling (Muennighoff et al., 2023).

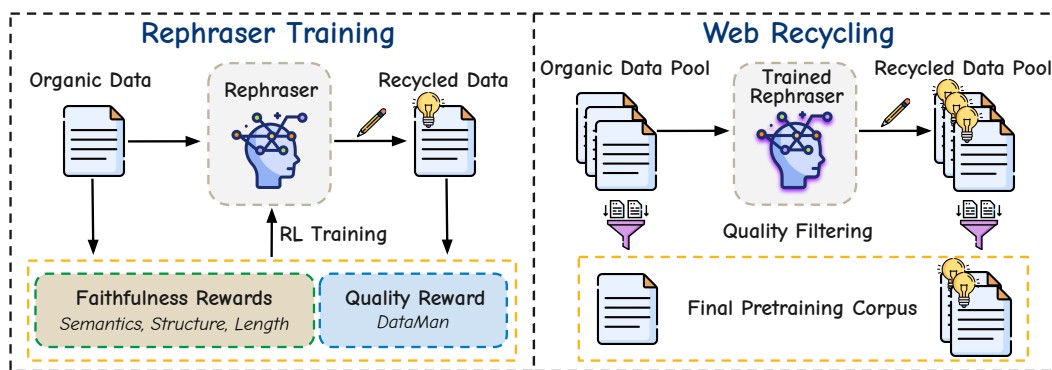

Figure 2: Overview of REPRO. We train our rephraser with quality and faithfulness rewards to optimize it to faithfully recycle the web.

As high-quality organic data becomes increasingly scarce, synthetic data emerges as a principled complement due to its flexibility and scalability (Havrilla et al., 2024). Recent work highlights the effectiveness of synthetic data in various training stages, spanning from pretraining (Maini et al., 2024), mid-training (Wang et al., 2025), to post-training (Li et al., 2025). In mid- and post-training, synthetic data generation typically leverages a set of high-quality seed data and employs LLMs to extract its essence or produce similar examples (Wang et al., 2023), which has been widely adopted to to incentivize math (Ge et al., 2024; Zhou et al., 2025b), reasoning (Wang et al., 2025), and instruction-following (Yue et al., 2024; Li et al., 2025) abilities. In contrast, synthetic data generation for pretraining is more challenging and less understood, confronting issues such as lack of diversity (Havrilla et al., 2024). Successful attempts, including Wikipedia-style rephrasing (Maini et al., 2024), guided rewriting (Nguyen et al., 2025), and topic-seeded generation (Li et al., 2023; Ben Allal et al., 2024; Hao et al., 2025), can enhance pretraining mixtures in both data quantity and quality (Su et al., 2025; Abdin et al., 2024; Maini et al., 2025).

Though promising, theory and practice caution that the indiscriminate use of synthetic data can trigger model collapse (Shumailov et al., 2024; Havrilla et al., 2024; Dohmatob et al., 2025), degrading generalization capabilities (Gerstgrasser et al., 2024; Feng et al., 2025). Specifically, Shumailov et al. (2024) show rapid degeneration when successive models are trained on their predecessors' outputs, with scaling-law changes explaining this as the tail-knowledge loss in synthetic data (Dohmatob et al., 2024). The distribution collapse issue is particularly risky in pretraining, as it not only degrades pretraining performance but harms more on the post-training outcomes (Chen et al., 2024). To prevent the potential collapse in pretraining, ProX (Zhou et al., 2025a) and RefineX (Bi et al., 2025) restrict the data synthesis output to conservative programs, refining pretraining data with a predefined set of operations, e.g., deletion and normalization.

## 3 METHODS

In this section, we present REPRO, an effective and faithful web recycling method for pretraining. We first introduce the web recycling setup (§3.1) and then our reinforcement learning approach to optimize an LM rephraser to faithfully recycle the web (§3.2). Our pipeline is illustrated in Figure 2.

### 3.1 RECYCLING PRETRAINING DATA

The construction of pretraining data starts from an organic data pool $\mathcal{D}_{\text{org}}$, obtained from web sources. As incorporating low-quality data into pretraining can significantly degrade model performance (Li et al., 2024), only high-quality samples are retained from $\mathcal{D}_{\text{org}}$. Formally, we define a quality function $Q(\cdot)$ with a threshold $\tau_{\text{org}}$ to select the high-quality subset:

$$\mathcal{D}_{\text{org-hq}} = \{x \in \mathcal{D}_{\text{org}} \mid Q(x) \geq \tau_{\text{org}}\}. \tag{1}$$

In data-limited scenarios (e.g., training frontier LLMs (Maini et al., 2025)), the total number of tokens in $\mathcal{D}_{\text{org-hq}}$, denoted by $B_{\text{org-hq}}$, is insufficient to cover the unique pretraining token budget $B$. To address this shortage, we introduce a language model rephraser $\mathcal{R}$, which transforms an organic data sample $x$ into a recycled data sample $x'$ given a rephrasing prompt $p$. All rephrasings then form

the recycled data pool $\mathcal{D}_{\text{rec}}$:

$$\mathcal{D}_{\text{rec}} = \{x' = \mathcal{R}(p, x) \mid x \in \mathcal{D}_{\text{org}}\}. \tag{2}$$

As with organic data, recycled samples are filtered by $Q(\cdot)$ to select the highest-quality portion:

$$\mathcal{D}_{\text{rec-hq}} = \{x \in \mathcal{D}_{\text{rec}} \mid Q(x) \geq \tau_{\text{rec}}\}, \tag{3}$$

where $\tau_{\text{rec}}$ is set so that the total number of tokens in $\mathcal{D}_{\text{rec-hq}}$ is $B - B_{\text{org-hq}}$.

The final pretraining dataset is then constructed from both organic and recycled high-quality subsets:

$$\mathcal{D}_{\text{final}} = \mathcal{D}_{\text{org-hq}} \cup \mathcal{D}_{\text{rec-hq}}. \tag{4}$$

## 3.2 TRAINING A FAITHFUL REPHRASER WITH RL

Table 1: Reward functions used in our RL for training an effective and faithful LM rephraser.

| Reward | Type | Functionality | Formal Definition |
|--------|------|---------------|-------------------|
| **DataMan** | Quality | Generate higher-quality data | DataMan(recycled) - DataMan(organic) |
| **BERTScore** | Faithfulness | Keep semantics consistent | BERTScore(organic, recycled) $\geq \tau_{\text{BERTScore}}$ |
| **Structure** | Faithfulness | Keep text structure consistent | Structure(organic, recycled) == 1 |
| **Length** | Faithfulness | Penalize free-form generation | Len(recycled) $\leq \tau_{\text{Length}} * $ Len(organic) |

To avoid model collapse when using synthetic data in pretraining (Chen et al., 2024), we leverage reinforcement learning (RL) to train a faithful rephraser for web recycling. Specifically, we design two types of rewards, *quality* and *faithfulness*, to guide RL optimization. The quality reward encourages the rephraser to produce outputs that are of higher quality than the organic data, while faithfulness rewards ensure that the rephrased data faithfully preserves the core semantic, structure, and length of the original text. The specific reward functions are as follows:

- **Data Quality (`DataMan`):** To encourage the generation of high-quality data, this reward incentivizes the rephraser to produce outputs $x'$ of better quality than the organic data $x$. Specifically, we choose DataMan (Peng et al., 2025), the state-of-the-art quality assessment method that prompts an LM to evaluate pretraining data across 13 criteria (e.g., coherence, topic focus, knowledge novelty) and assign an overall score.

$$r_{\text{DataMan}}(x, x') = \text{DataMan}(x') - \text{DataMan}(x) \tag{5}$$

- **Semantic Faithfulness (`BERTScore`):** To ensure that the rephrased text preserves the core meaning of the organic data, we reward semantic similarity between $x$ and $x'$ when their BERTScore (Zhang et al., 2020) is higher than a preset threshold $\tau_{\text{BERTScore}}$.

$$r_{\text{BERTScore}}(x, x') = \mathbf{1}[\text{BERTScore}(x, x') \geq \tau_{\text{BERTScore}}] \tag{6}$$

- **Structural Faithfulness (`Structure`):** To encourage the rephraser to maintain the high-level textual structure (e.g., Markdown) of the organic data and prevent the loss of the structural diversity, we apply a structure comparison LM to assess structural faithfulness by in-context learning.

$$r_{\text{Structure}}(x, x') = \mathbf{1}[\text{Structure}(x, x') = 1] \tag{7}$$

- **Length Faithfulness (`Length`):** To further penalize uncontrolled or free-form generation that deviates significantly from the organic data, we impose a constraint on the length of the recycled text relative to the original by a factor of $\tau_{\text{Length}}$.

$$r_{\text{Length}}(x, x') = \mathbf{1}[\text{Len}(x') \leq \tau_{\text{Length}} * \text{Len}(x)] \tag{8}$$

The final reward function $r(x, x')$ combines these individual rewards with weighting coefficients $\lambda$:

$$r(x, x') = \lambda_{\text{DataMan}} r_{\text{DataMan}} + \lambda_{\text{BERTScore}} r_{\text{BERTScore}} + \lambda_{\text{Structure}} r_{\text{Structure}} + \lambda_{\text{Length}} r_{\text{Length}}. \tag{9}$$

We employ Group Relative Policy Optimization (GRPO) algorithm (Shao et al., 2024) for RL training, starting from the base model $\mathcal{R}_{\text{base}}$ with organic data $x$ sampled from $\mathcal{D}_{\text{grpo}} \subset \mathcal{D}_{\text{org}}$. GRPO

enhances training stability by normalizing advantage estimates from a group of $n$ generated outputs $\{x'_1, \ldots, x'_n\}$. The advantage of $x'_i$ is used to update the policy via a clipped surrogate objective:

$$\mathcal{R}_{\text{rl}} = \arg\max_{\mathcal{R}} \mathbb{E}_{\substack{x \sim \mathcal{D}_{\text{grpo}}, \\ x'_i \sim \pi_{\mathcal{R}}}} \left[ \min\left( P \hat{A}_i, \text{clip}\left( P, 1 - \epsilon, 1 + \epsilon \right) \hat{A}_i \right) - \beta \mathbb{D}_{\text{KL}}(\mathcal{R} \| \mathcal{R}_{\text{base}}) \right], \quad (10)$$

$$\text{where } P = \frac{\pi_{\mathcal{R}}(x'_i \mid p, x)}{\pi_{\mathcal{R}_{\text{base}}}(x'_i \mid p, x)}, \hat{A}_i = \frac{r(x, x'_i) - \text{mean}(\mathbf{r})}{\text{std}(\mathbf{r})}, \text{and } \mathbf{r} = \{r(x, x'_1), \ldots, r(x, x'_n)\}. \quad (11)$$

$\pi_{\mathcal{R}}(x' \mid p, x)$ denotes the probability of the rephraser model generating $x'$ from $x$. The final optimized model $\mathcal{R}_{\text{rl}}$ generates the recycled data pool $\mathcal{D}_{\text{rec}}$ from the organic data pool $\mathcal{D}_{\text{org}}$, which then constitutes the pretraining dataset following the steps in Section 3.1.

## 4 EXPERIMENTAL SETUP

**Pretraining Data and Model.** We conduct our main experiments on the DCLM dataset (Li et al., 2024). Specifically, we follow ReWire (Nguyen et al., 2025) and randomly sample 72B/144B tokens from DCLM-RefinedWeb as our organic data pool $\mathcal{D}_{\text{org}}$. DCLM-RefinedWeb applies rule-based filtering and global deduplication to Common Crawl (Penedo et al., 2023), but not model-based filtering, making it a moderate-quality data source well suited for recycling. The unique token budget $B$ is 14.4B/28.8B in our main experiment, and we explore different budgets in later analyses. The quality function $Q$ is DCLM-fastText, the best-performing classifier in data selection from DCLM-RefinedWeb. Following the quality filtering threshold $\tau_{\text{org}}$=0.018112 in DCLM, the amount of high-quality organic data $B_{\text{org-hq}}$ is 7.2B/14.4B. For our pretraining models, we use a decoder-only architecture and adopt three commonly used parameter scales (400M, 1.4B, and 2.8B) from previous works (Yu et al., 2025; Nguyen et al., 2025), and train all models from scratch.

**Evaluation.** We evaluate pretrained models on 22 downstream tasks in either zero-shot or few-shot manners. These tasks provide a comprehensive assessment of essential model abilities, including commonsense reasoning, language understanding, reading comprehension, symbolic problem solving, and world knowledge. We use centered accuracy as the primary evaluation metric, where accuracy per task is mapped to 0 for random guessing and 1 for perfect accuracy. The average centered accuracy across all tasks is denoted as "Core score". Following Yu et al. (2025), we exclude CommonsenseQA from the original DCLM-Core due to its high instability and limited informativeness. Full task details and their few-shot numbers are provided in Appendix C.1.

**Baselines.** We compare REPRO with (1) *organic data only*; (2) *WRAP* (Maini et al., 2024): Wikipedia-style rephrasing using Mistral-7B-Instruct-v0.1 (Jiang et al., 2023); (3) *ProX* (Zhou et al., 2025a): fine-grained edits of organic data via model-generated programs; and (4) *ReWire* (Nguyen et al., 2025): guided rewriting with chain-of-thought reasoning using Llama-3.3-70B-Instruct (Dubey et al., 2024). As ReWire's code has not been open-sourced, we randomly sample 7.2B tokens from its released data. This may give ReWire a performance advantage, since it essentially utilizes a larger pool (our organic + their organic). These baselines cover state-of-the-art web recycling techniques such as prompting, program-based editing, and chain-of-thought reasoning. Some concurrent works, such as RefineX (Bi et al., 2025) and BeyondWeb (Maini et al., 2025), have not open-sourced their data and code, so we leave the comparison with them in the future. More baseline details can be found in Appendix C.2.

**Implementation Details.** We initialize our rephraser model $\mathcal{R}$ by Qwen3-4B (Yang et al., 2025). In RL, we set the cutting threshold $\tau_{\text{BERTScore}}$ as 0.65, and set $\tau_{\text{Length}}$ as 1.25. We utilize released DataMan (1.5B) and BERTScore (350M) models for their reward calculation and construct our structure comparison model by prompting Qwen3-4B with few-shot examples. The coefficients $\lambda_{\text{DataMan}}, \lambda_{\text{BERTScore}}, \lambda_{\text{Structure}}$, and $\lambda_{\text{Length}}$ are set to 3, 1, 1, 1, respectively. The size of the RL dataset $\mathcal{D}_{\text{grpo}}$ is 41k. $\mathcal{D}_{\text{grpo}}$ consists solely of organic data with a DataMan score below 5, since 5 is the maximum score and thus cannot be improved. The clipping $\epsilon$ in GRPO is 0.2, $\beta$ is 0.005, and the number of rollouts $n$ per input is 8. To examine whether a strong prior can improve RL, we also add an **optional** supervised fine-tuning (SFT) stage before RL, where we use GPT-4o (Achiam et al., 2023) as the teacher to generate 50k example rephrasings to warm up our rephraser. Unless otherwise stated, REPRO denotes **RL w/o SFT**. In the inference stage, we adopt vllm (Kwon et al., 2023) framework for efficient text generation using our rephraser. Following Nguyen et al. (2025),

Table 2: Benchmarking different web recycling methods on DCLM with 400M, 1.4B, and 2.8B pretraining models. **Bold** and underline indicate the best and second-best results.

| Method | Pool | Unique Data | Commonsense Reasoning (3 tasks) | Language Understanding (6 tasks) | Reading Comprehension (3 tasks) | Symbolic Problem (5 tasks) | World Knowledge (5 tasks) | Core (22 tasks) |
|---|---|---|---|---|---|---|---|---|
| **400M Setting:** 400M model, 28.8B training tokens | | | | | | | | |
| Organic | 72B | 7.2B | 0.23613 | 0.27079 | 0.03724 | 0.14535 | 0.20126 | 0.18990 |
| Organic | 72B | 14.4B | 0.26953 | 0.25781 | 0.05623 | 0.14991 | 0.17683 | 0.18899 |
| WRAP | 72B | 7.2B + 7.2B | 0.24784 | 0.25798 | 0.06269 | 0.16303 | 0.20067 | 0.19536 |
| ProX | 72B | 7.2B + 7.2B | 0.24252 | 0.25403 | 0.06884 | 0.16528 | 0.20647 | 0.19623 |
| ReWire | 72B | 7.2B + 7.2B | 0.24051 | 0.26453 | 0.06232 | 0.17392 | **0.21246** | 0.20125 |
| REPRO | 72B | 7.2B + 7.2B | **0.28454** | **0.27792** | **0.07181** | **0.19409** | 0.21154 | **0.21658** |
| **1B Setting:** 1.4B model, 28.8B training tokens | | | | | | | | |
| Organic | 72B | 7.2B | 0.32348 | 0.38371 | 0.19584 | 0.19795 | 0.28745 | 0.28578 |
| Organic | 72B | 14.4B | 0.31483 | 0.37916 | 0.15112 | 0.17507 | 0.28311 | 0.27108 |
| WRAP | 72B | 7.2B + 7.2B | 0.33860 | 0.36873 | 0.20071 | 0.20492 | 0.27576 | 0.28335 |
| ProX | 72B | 7.2B + 7.2B | 0.34246 | 0.38253 | 0.21735 | 0.18487 | 0.29641 | 0.29004 |
| ReWire | 72B | 7.2B + 7.2B | 0.33330 | 0.37400 | **0.23369** | 0.17641 | **0.31186** | 0.29029 |
| REPRO | 72B | 7.2B + 7.2B | **0.36776** | **0.38519** | 0.20832 | **0.20597** | 0.30304 | **0.29929** |
| **3B Setting:** 2.8B model, 55.9B training tokens | | | | | | | | |
| Organic | 144B | 14.4B + 14.4B | 0.41490 | 0.47105 | 0.30908 | 0.19259 | 0.36491 | 0.35390 |
| WRAP | 144B | 14.4B + 14.4B | 0.41334 | 0.44467 | 0.27998 | 0.19308 | 0.35327 | 0.33999 |
| ProX | 144B | 14.4B + 14.4B | 0.37050 | 0.47198 | 0.31033 | 0.19441 | 0.37112 | 0.35688 |
| ReWire | 144B | 14.4B + 14.4B | 0.40759 | 0.47302 | 0.31280 | 0.19138 | 0.37658 | 0.35632 |
| REPRO | 144B | 14.4B + 14.4B | 0.41768 | 0.47535 | 0.32152 | 0.20401 | **0.37801** | 0.36272 |
| REPRO | 144B | 14.4B + 7.2B | **0.45065** | **0.48314** | **0.33544** | **0.20831** | 0.37045 | **0.37050** |

we use a temperature of 1.0 and a top-p of 0.9. We conduct a key hyperparameter analysis in Appendix B.3 and provide all prompts used in our experiments in Appendix C.4.

## 5 EVALUATION RESULTS

In this section, we present our main results on DCLM, along with different unique token budgets (§5.1). Then, we perform ablation studies (§5.2), validate the recycling of non-high-quality data (§5.3), dive into the RL training dynamics with different reward choices (§5.4), and analyze distributional features of recycled data (§5.5). We conclude with an in-depth analysis of the operations performed by our rephraser (§5.6). Additional results and ablations can be found in Appendix B.

### 5.1 MAIN RESULTS

**Overall Performance.** Table 2 demonstrates the overall performance of different web recycling methods on DCLM-RefinedWeb. First, using top-10% high-quality organic data outperforms using top-20%, confirming the necessity of quality filtering. Our method, REPRO, significantly outperforms using organic data only, achieving 4.7%-14.0% relative improvements in Core scores. This indicates that our recycled data can effectively complement organic data in the data-limited setup to enhance the pretraining performance of LLMs. Our method also consistently **doubles the performance gains** achieved by all baselines across different pretraining model scales, including the one (ReWire) that prompts a much larger model (Llama-3.3-70B-Instruct) to rephrase the corpus. We would like to note that, compared to the 400M setup, the 3B setup is undertrained due to our compute constraints, which may limit the absolute performance gains across all methods. We also observe that in the 3B setup, Wikipedia-style rephrasing (WRAP) can even harm performance, likely due to its limited diversity and weaker preservation of the original distribution, both of which become increasingly important for larger pretraining models. Furthermore, we provide the efficiency advantage of REPRO in Appendix B.1, where it achieves a $36.7\times$ speedup compared to ReWire. In summary, these results highlight that even a relatively small (4B) model can learn without the guidance of an LLM to be an effective rephraser to generate high-quality recycled data.

Table 3: Ablation studies on different components of REPRO in the 400M setup. Prompting$^+$ and SFT$^+$ denote the use of an enhanced prompt (Appendix C.4) with reward descriptions.

| Method | Pool | Unique Data | Commonsense Reasoning (3 tasks) | Language Understanding (6 tasks) | Reading Comprehension (3 tasks) | Symbolic Problem (5 tasks) | World Knowledge (5 tasks) | Core (22 tasks) |
|---|---|---|---|---|---|---|---|---|
| **400M Setting:** 400M model, 28.8B training tokens | | | | | | | | |
| Organic | 72B | 14.4B | 0.26953 | 0.25781 | 0.05623 | 0.14991 | 0.17683 | 0.18899 |
| Prompting | 72B | 7.2B + 7.2B | 0.24310 | 0.26758 | 0.05075 | 0.17392 | 0.20196 | 0.19847 |
| Prompting$^+$ | 72B | 7.2B + 7.2B | 0.27831 | 0.25229 | 0.06870 | 0.16475 | 0.20032 | 0.19910 |
| SFT | 72B | 7.2B + 7.2B | 0.24447 | 0.24920 | 0.04013 | 0.16564 | 0.21009 | 0.19216 |
| SFT$^+$ | 72B | 7.2B + 7.2B | 0.26049 | 0.27111 | 0.06593 | 0.16899 | 0.20206 | 0.20278 |
| RL w/o Faithfulness | 72B | 7.2B + 7.2B | 0.22357 | 0.25633 | 0.06822 | 0.16961 | 0.20376 | 0.19456 |
| SFT + Full RL | 72B | 7.2B + 7.2B | **0.29210** | 0.25965 | 0.07141 | 0.16988 | **0.21632** | 0.20816 |
| REPRO (Full RL) | 72B | 7.2B + 7.2B | 0.28454 | **0.27792** | **0.07181** | **0.19409** | 0.21154 | **0.21658** |

**Improving Organic Data Efficiency.** We also evaluate the effectiveness of REPRO under different unique token budgets $B$. Specifically, we vary $B$ from 14.4B (our main experiment) to 21.6B and 28.8B (our total training budget) in the 1B setup. As shown in Figure 3, REPRO consistently outperforms all baselines across different $B$ by a large margin. Among these budgets, the best performance of REPRO is achieved at $B$=14.4B, and $B$=21.6B yields comparable performance but delivers the largest improvement over baseline methods. This implies that REPRO can potentially improve organic data efficiency by 2-3×. However, further increasing $B$ to 28.8B leads to a performance drop, likely due to the inclusion of more moderate-to-low quality data, which reduces the advantage of higher quality recycled tokens. Overall, these results highlight the effectiveness of our method in increasing the amount of high-quality pretraining data.

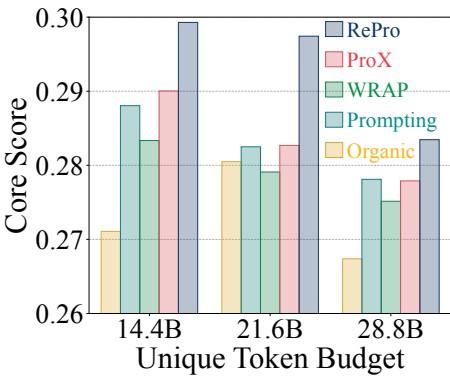

Figure 3: Performance of recycling methods w.r.t. different unique token budgets in the 1B setup.

## 5.2 ABLATION STUDIES

We conduct ablation studies in the 400M setup to investigate the effectiveness of each component in our rephraser training. Comparisons include our base rephraser (Qwen3-4B) with direct prompting, the rephraser after SFT using GPT-4o rephrasings, and the rephraser from RL training without faithfulness rewards. Note that we do not include the model trained without the quality reward, as in this case, the rephraser would simply learn to copy the organic data to maximize faithfulness rewards. As compared in Table 3, straightforward prompting or SFT alone could not greatly benefit overall performance compared to the organic baseline. This implies a significant performance gap between prompting-based rephrasers (even as strong as GPT-4o) and our training-based rephraser, which has been specifically optimized for generating better recycled data.

With SFT prior, RL may achieve further gains on world knowledge tasks but falls behind in the overall Core score. We hypothesize that the SFT data generated by GPT-4o may contain distillation-style rephrasings that benefit knowledge-intensive tasks but hinder generalization to others, a common issue noted in previous works (Su et al., 2025; Maini et al., 2025). This comparison highlights that REPRO does not rely on supervised signals from an external LLM to perform effective recycling. Removing faithfulness rewards from RL leads to a substantial performance drop in Core score, indicating that the faithfulness of recycled data is also crucial to the effectiveness of web recycling.

To make the comparison with our method more informative, we also strengthen prompting and SFT baselines by adding the descriptions of our reward functions directly into the prompt, denoted as Prompting$^+$ and SFT$^+$. The results show that Prompting$^+$ yields only a negligible improvement over our base prompt, and SFT$^+$, although much better than SFT, still remains clearly below REPRO. Overall, these results demonstrate that a dedicated RL procedure is more effective than prompting alone for shaping the base model into a faithful and high-performing rephraser.

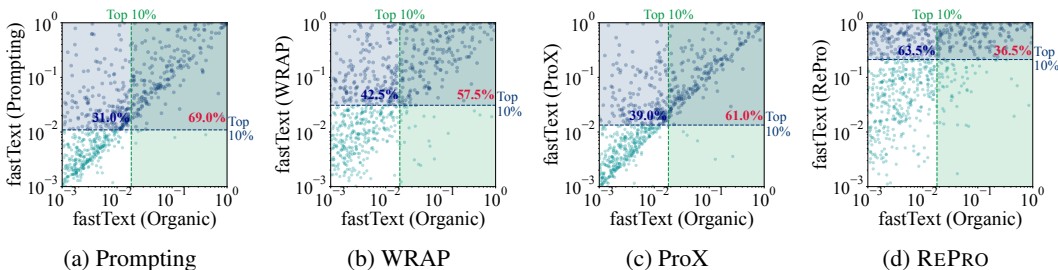

(a) Prompting      (b) WRAP      (c) ProX      (d) REPRO

Figure 4: Relationship between the fastText scores before and after rephrasing with each method.

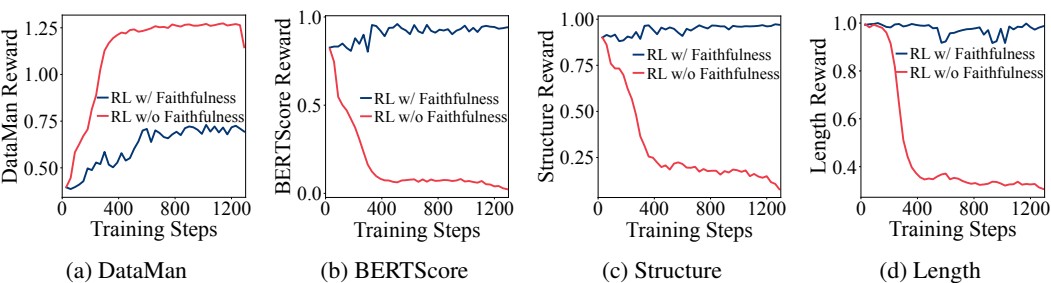

(a) DataMan      (b) BERTScore      (c) Structure      (d) Length

Figure 5: Validation curves of (a) DataMan, (b) BERTScore, (c) structure, and (d) length rewards during our RL training with and without faithfulness rewards.

## 5.3 RECYCLING OF NON-HIGH-QUALITY DATA

In this experiment, we measure how much of the high-quality recycled data $\mathcal{D}_{\text{rec-hq}}$ comes from non-high-quality organic data $\mathcal{D}_{\text{org}} \setminus \mathcal{D}_{\text{org-hq}}$. We exclude ReWire from this comparison because its high-quality cutoff threshold is not publicly available. As illustrated in Figure 4, for REPRO, almost **two-thirds (63.5%)** of $\mathcal{D}_{\text{rec-hq}}$ originates from $\mathcal{D}_{\text{org}} \setminus \mathcal{D}_{\text{org-hq}}$. In comparison, the corresponding proportions for Prompting, WRAP, and ProX are 31.0%, 42.5%, and 39.0%, respectively, indicating that REPRO converts substantially more low-quality data into high-quality rephrasings. Furthermore, we observe that REPRO adopts a higher cutoff threshold for high-quality data than other methods, which suggests that our rephraser achieves a larger improvement in the overall fastText scores. This further explains why REPRO demonstrates better organic data efficiency in Figure 3, as the overall quality of its rephrased data is higher than that of the other methods.

To better understand how REPRO operates on different data quality, we provide more recycling cases in Appendix D.1. Thanks to the RL optimization, REPRO preserves the key characteristics of organic data (writing style, structure, genre, etc.) and learns to flexibly apply various operations to polish the text. For instance, REPRO often filters out noisy or irrelevant content, clarifies unclear information, and refines the organization and writing of already high-quality texts to further enhance their learnability. In summary, our method effectively transforms diverse data into higher-quality forms, enabling more efficient use of organic data.

## 5.4 EFFECTIVENESS OF REWARD FUNCTIONS

This set of experiments studies the effect of reward functions. In Figure 5, we plot the training dynamics of our RL training with and without faithfulness rewards. The rewards shown are calculated on a validation set of 128 randomly sampled data from $\mathcal{D}_{\text{grpo}}$, using mean value as the aggregation function. We observe that with all rewards enabled, DataMan, BERTScore, and structure rewards all steadily improve during training (Figures 5a, 5b, and 5c), while the length reward remains stably high (Figure 5d). This indicates that these four rewards can be effectively optimized together (instead of trade-offs) to improve the capability of our rephraser to generate high-quality and consistent recycled data. In contrast, without faithfulness rewards as constraints (by setting their coefficients to zero), DataMan reward quickly converges to a high value (Figure 5a), but all other three rewards drop significantly (Figures 5b, 5c, and 5d).

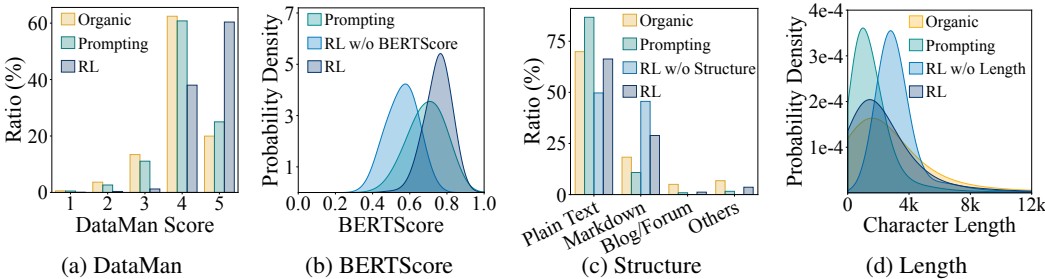

|        |        |        |        |
|--------|--------|--------|--------|
| (a) DataMan | (b) BERTScore | (c) Structure | (d) Length |

Figure 7: Reward-related feature distributions of 30,000 recycled data generated by different rephrasers. Features are (a) DataMan score, (b) BERTScore, (c) structure type, and (d) text length.

We further explore the impact of using other quality rewards in our RL training. Specifically, we investigate two additional quality reward options: (1) DCLM-fastText, a n-gram-based classifier that performs the best in data selection from DCLM-RefinedWeb, and (2) training data influence (Yu et al., 2024), which measures the actual training effect of each data point on the reference task given a pretrained model checkpoint. Following Yu et al. (2025), we use our 400M organic baseline at 10k steps as the model checkpoint and adopt FLAN (Wei et al., 2022) as the reference task.

As illustrated in Figure 6, the fastText reward quickly saturates at a high value, while the influence reward remains low and drops sharply to 0 (no rephrasing) at the end. A case study in Appendix D.2 shows that the rephraser trained with fastText reward tends to generate text with an academic tone, which hacks the n-gram-based classifier to get high scores. In contrast, the rephraser trained with the influence reward struggles to explore rephrasing strategies that consistently improve the reward, likely because this signal is too fine-grained and difficult to optimize upon. Our findings suggest that a useful metric for data selection may not be an appropriate reward for rephraser RL. A good quality reward should not only correlate well with the final pretraining performance, but also be robust to reward hacking and can be effectively optimized.

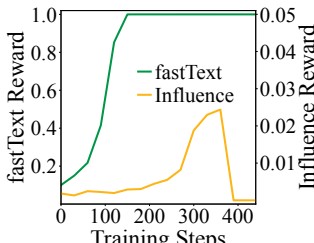

Figure 6: RL training with DCLM-fastText or data influence as quality rewards.

## 5.5 DISTRIBUTIONAL ANALYSES

This set of experiments examines the effectiveness of each reward function used in our RL training from a distributional perspective. To analyze each reward function, we plot the distributions of relevant features for organic data alongside data generated by different rephrasers. Comparisons include our base rephraser (Qwen3-4B) with direct prompting, the rephraser trained without a specific reward, and our final rephraser trained with all rewards. All features are calculated on a set of 30k randomly sampled instances from our data pool.

First, we compare the distributions of the DataMan score in Figure 7a. REPRO effectively shifts the distribution towards higher DataMan scores, boosting the proportion of data with a Score of 5 from 20% to **60%**. In contrast, direct prompting yields a marginal improvement over organic data, with only 25% of recycled data achieving a Score of 5.

With faithfulness rewards, REPRO successfully preserves the original characteristics at the distribution level. As shown in Figure 7b, REPRO maintains semantic similarity with an average BERTScore of **0.75**, compared to 0.69 for direct prompting and 0.56 for RL without the BERTScore reward, underscoring the necessity of this reward for semantic preservation.

We further assess structural diversity by prompting *Gemini 2.5 Flash-Lite* to classify the text structure as plain text, Markdown, blog/forum, or others. As shown in Figure 7c, direct prompting tends to transform Markdown-style text into plain text, whereas RL without the structure reward overgenerates Markdown-style text, harming structural diversity. The structural distribution of our final recycled data aligns most closely with that of organic data.

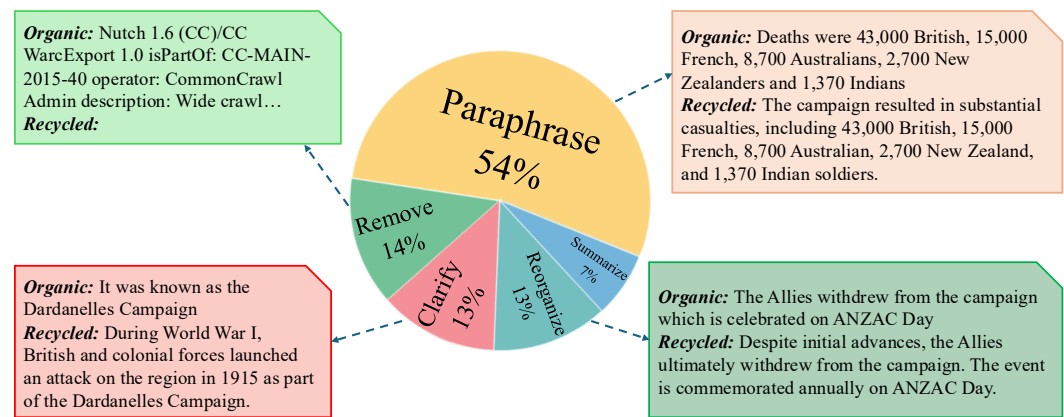

*Organic:* Nutch 1.6 (CC)/CC WarcExport 1.0 isPartOf: CC-MAIN-2015-40 operator: CommonCrawl Admin description: Wide crawl…
*Recycled:*

*Organic:* Deaths were 43,000 British, 15,000 French, 8,700 Australians, 2,700 New Zealanders and 1,370 Indians
*Recycled:* The campaign resulted in substantial casualties, including 43,000 British, 15,000 French, 8,700 Australian, 2,700 New Zealand, and 1,370 Indian soldiers.

*Organic:* It was known as the Dardanelles Campaign
*Recycled:* During World War I, British and colonial forces launched an attack on the region in 1915 as part of the Dardanelles Campaign.

*Organic:* The Allies withdrew from the campaign which is celebrated on ANZAC Day
*Recycled:* Despite initial advances, the Allies ultimately withdrew from the campaign. The event is commemorated annually on ANZAC Day.

Figure 8: Operations performed by REPRO and their corresponding cases.

Similarly, for text length (Figure 7d), we observe that direct prompting tends to produce shorter, summary-like text, while RL without the length reward generates much longer text that may introduce extraneous information (see Appendix D.2 for more details). Once again, our recycled data aligns most closely with the length distribution of the organic data.

In summary, these results confirm the effectiveness of each reward function in our RL framework. They coordinate together to simultaneously improve data quality and shape the overall distribution of recycled data to mirror that of organic data, enabling an effective and faithful recycling process.

## 5.6 REPHRASING OPERATIONS

Finally, we analyze the specific operations performed by our rephraser to better understand its behaviors. We randomly sample 100 instances from our recycled data pool and again prompt *Gemini 2.5 Flash-Lite* to classify the operations (verb + noun) performed by our rephraser. We categorize the extracted operations into 5 primary types: *paraphrasing*, *removing*, *clarification*, *reorganization*, and *summarization*. The distributions and examples of each operation are illustrated in Figure 8. We observe that paraphrasing is the most popular operation, which can improve writing quality by fixing grammatical errors and enhancing fluency. Removing is the second common one, which eliminates irrelevant content such as advertisements and metadata. Other operations like clarification, reorganization, and summarization are also frequently used to enhance the informativeness and coherence of the text. Overall, these results demonstrate that our rephraser learns to flexibly apply a variety of operations (instead of a manually defined set) to effectively recycle web data into high-quality training data while respecting the original content.

## 6 CONCLUSION

In this paper, we introduce REPRO, a novel web recycling method that trains a 4B language model to generate high-quality and faithful rephrasings of web data. Pretraining results show that RE-PRO achieves 4.7%-14.0% relative accuracy improvements over organic-only baselines, outperforms state-of-the-art recycling method ReWire, which prompts a 70B model, and even surpasses an organic baseline whose data pool is enlarged by $4\times$. Distributional analyses confirm that REPRO maintains the semantic meaning, structural diversity, and length balance of organic data.

In conclusion, our research provides two key takeaways for effective web recycling. First, rephrasing does not necessarily require a large language model; small models trained with carefully designed rewards can recycle the web more effectively and faithfully. Second, preserving the essential characteristics of organic data not only helps maintain the original distribution but also enhances pretraining performance with recycled data. Future work can explore more diverse and verifiable reward signals, such as a checklist, to further improve the quality and faithfulness of recycled data. We hope that our work inspires more cost-efficient and reliable approaches to alleviate the data wall and enable sustainable scaling of LLM pretraining.

## REPRODUCIBILITY STATEMENT

We have made extensive efforts to ensure the reproducibility of our results. Anonymized source code is available at https://anonymous.4open.science/r/RePro to allow independent verification of our implementation. Our experimental setup is described in detail in Section 4, including the dataset used, model architectures, training and inference pipelines, and hyperparameters in our method to facilitate faithful reproduction of our experiments. Additional implementation details are included in Appendix C, and we reference all external resources used in our work. These materials provide the necessary information for others to reproduce and build upon our results.

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

## A   Use of Large Language Models in Paper Writing

We use LLMs to assist with writing and language refinement, specifically to improve grammar, phrasing, and clarity. We also use them to support our literature search and review. All methodological content, experiments, analysis, and interpretation were carried out solely by the authors. The use of LLMs was strictly limited to text polishing and literature review.

## B   Additional Results

This section presents additional results that complement our main findings.

### B.1   Efficiency Comparison

We provide a detailed comparison of the training and inference hours required for each rephrasing method in Table 4. REPRO demonstrates significant efficiency advantages over other approaches, with a 1.2× speedup compared to WRAP and a 36.7× speedup compared to ReWire. This confirms that our recycling method is not only more effective but also more cost-efficient, making it a practical choice for large-scale pretraining.

Table 4: H100 hours of rephrasing 72B tokens. We exclude ProX, as it is an operation-based method rather than rephrasing.

|  | WRAP | ReWire | REPRO |
|---|---|---|---|
| **Training** | - | - | 192 |
| **Inference** | 2,095 | 63,360 | 1,536 |
| **Total** | 2,095 | 63,360 | 1,728 |

### B.2   Improvements on All DataMan Criteria

In this section, we present the improvements of REPRO on all DataMan criteria. As shown in Figure 9, despite only being optimized for the overall score, REPRO consistently outperforms the organic data and prompting baseline across all individual criteria. For some subjective criteria like *knowledge novelty* and *creativity*, the ratio of Score=5 remains low after rephrasing. This is expected, as these aspects are inherently challenging to enhance through rephrasing alone, especially given our rephraser is trained to faithfully preserve the original content. In summary, these results highlight the effectiveness of REPRO in improving various dimensions of data quality.

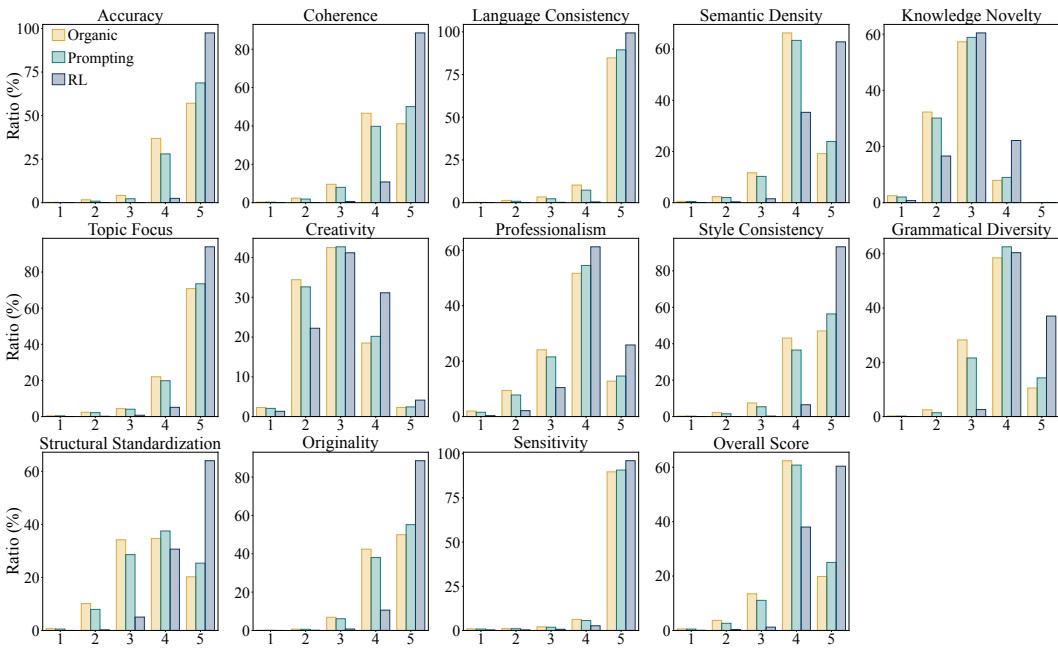

Figure 9: Improvements of REPRO on all DataMan criteria.

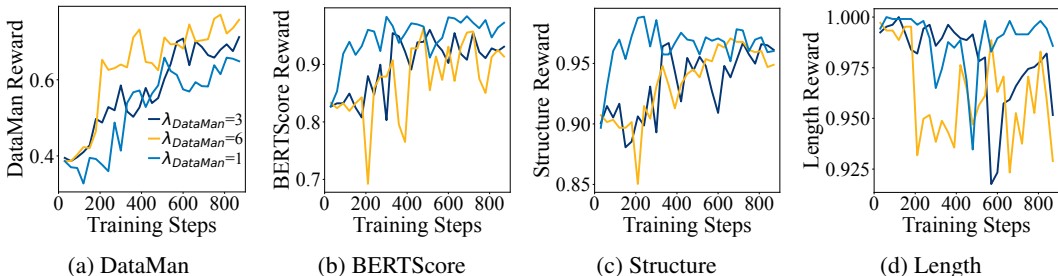

(a) DataMan  (b) BERTScore  (c) Structure  (d) Length

Figure 10: Validation curves of (a) DataMan, (b) BERTScore, (c) structure, and (d) length rewards during our RL training with different DataMan coefficients.

Table 5: Data selection with different quality functions.

| Quality Function | Pool | Unique Data | Commonsense Reasoning (3 tasks) | Language Understanding (6 tasks) | Reading Comprehension (3 tasks) | Symbolic Problem (5 tasks) | World Knowledge (5 tasks) | Core (22 tasks) |
|---|---|---|---|---|---|---|---|---|
| **1B Setting:** 1.4B model, 28.8B training tokens | | | | | | | | |
| Random | 72B | 7.2B | 0.28256 | 0.33930 | 0.19857 | 0.16795 | 0.23611 | 0.24998 |
| DCLM-fastText | 72B | 7.2B | 0.32348 | **0.38371** | 0.19584 | **0.19795** | **0.28745** | **0.28578** |
| DCLM-fastText | 72B | 14.4B | 0.31483 | 0.37916 | 0.15112 | 0.17507 | 0.28311 | 0.27108 |
| DataMan | 72B | 7.2B | 0.31369 | 0.32591 | 0.17524 | 0.16676 | 0.27709 | 0.25643 |
| DCLM-fastText + DataMan | 72B | 7.2B + 7.2B | **0.32643** | 0.37210 | **0.19902** | 0.19010 | 0.28630 | 0.28141 |

### B.3 HYPERPARAMETER STUDIES

In this study, we vary the coefficient of DataMan reward $\lambda_{\text{DataMan}}$ to 1, 3 (our main setup), and 6. As shown in Figure 10, all reward curves increase steadily and show a very similar pattern, showing the robustness of our method. Upweighting DataMan reward can make it increase faster, yet at the cost of larger fluctuations in faithfulness rewards, and vice versa. Our final choice strikes a decent balance between DataMan and faithfulness rewards. Our compute resources do not support a larger RL, e.g., training Qwen3-8B, while concurrent work (Maini et al., 2025) finds a diminishing return when using an 8B rephraser compared to 3B.

For $\tau_{\text{BERTScore}}$, we set it as 0.65, since we find that about two-thirds of the recycled data by prompting can achieve a BERTScore above this value. It is a representative cutoff that reflects the majority semantic similarity level of the recycled data while still leaving room to improve rephrasing quality.

### B.4 DATAMAN AS QUALITY FUNCTION

In this section, we evaluate the effectiveness of using DataMan as a quality function for selecting organic data. As shown in Table 5, either using DataMan selection only or combining it with DCLM-fastText selection could not beat using DCLM-fastText alone. This suggests that (1) simply merging multiple quality functions may not enlarge the amount of high-quality data, highlighting the importance of recycling, and (2) the reward in rephraser training is not necessarily the best quality function for data selection, which strengthens our findings in Section 5.4.

## C ADDITIONAL EXPERIMENT DETAILS

This section provides additional details about our experiments.

### C.1 EVALUATION TASKS

We list all evaluation task names and their few-shot numbers in Table 6.

### C.2 BASELINES

We provide the implementation details of the baselines we compare against in our main results:

Table 6: All evaluation task names and their few-shot numbers.

| Category | Task | #Shots |
|---|---|---|
| Commonsense Reasoning | copa (Roemmele et al., 2011) | 0 |
| | openbook_qa (Mihaylov et al., 2018) | 0 |
| | piqa (Bisk et al., 2020) | 10 |
| Language Understanding | bigbench_language_identification (Srivastava et al., 2023) | 10 |
| | hellaswag_zeroshot (Zellers et al., 2019) | 0 |
| | hellaswag (Zellers et al., 2019) | 10 |
| | lambada_openai (Paperno et al., 2016) | 0 |
| | winograd (Levesque et al., 2012) | 0 |
| | winogrande (Sakaguchi et al., 2020) | 0 |
| Reading Comprehension | boolq (Clark et al., 2019) | 10 |
| | coqa (Reddy et al., 2019) | 0 |
| | squad (Rajpurkar et al., 2016) | 10 |
| Symbolic Problem | agi_eval_lsat_ar (Zhong et al., 2024) | 3 |
| | bigbench_cs_algorithms (Srivastava et al., 2023) | 10 |
| | bigbench_dyck_languages (Srivastava et al., 2023) | 10 |
| | bigbench_operators (Srivastava et al., 2023) | 10 |
| | bigbench_repeat_copy_logic (Srivastava et al., 2023) | 10 |
| World Knowledge | arc_challenge (Clark et al., 2018) | 10 |
| | arc_easy (Clark et al., 2018) | 10 |
| | bigbench_qa_wikidata (Srivastava et al., 2023) | 10 |
| | jeopardy (Tunguz, 2019) | 10 |
| | mmlu_fewshot (Hendrycks et al., 2021) | 5 |

- WRAP (Maini et al., 2024): Following their original paper, we adopt Mistral-7B-Instruct-v0.1 (Jiang et al., 2023) as the rephraser, using their Wikipedia-style prompt.

- ProX (Zhou et al., 2025a): This method uses Llama-3-70B-Instruct (Dubey et al., 2024) to annotate chunk-level programs to polish the organic data. Then, they train a 0.3B model to learn the annotated programs and perform final edits. We emphasize that their recycling mechanism relies on program-based operations, in contrast to our generation-based rephrasing. These two mechanisms are essentially orthogonal, and we believe future work can leverage both to more effectively recycle the web.

- ReWire (Nguyen et al., 2025): This method employs guided rewriting with chain-of-thought reasoning using Llama-3.3-70B-Instruct (Dubey et al., 2024). As their code has not been open-sourced, we randomly sample 7.2B tokens from their released data. This may give them a performance advantage, since their organic pool differs from ours, meaning their method in our main results actually utilizes a larger pool.

## C.3 TRAINING HYPERPARAMETERS

We provide our training hyperparameters in Table 7.

Table 7: Training details.

| Hyperparameter | 400M LM | 1.4B LM | Rephraser |
|---|---|---|---|
| Steps | 27462 | 54923 | 2000 |
| Batch size | 512 | 256 | 24 |
| Sequence length | 2048 | 2048 | 4096 (2048i + 2048o) |
| Max learning rate | 3e-3 | 3e-3 | 1e-6 |
| Optimizer | AdamW | AdamW | AdamW |
| Scheduler | Cosine | Cosine | Cosine |

## C.4 PROMPTS

---

**REPRO Prompt**

Your task is to read and paraphrase the provided text following these instructions:
- Delete clearly irrelevant content:
  - Website headers, navigation bars, or menu items (e.g., "Home — About — Contact")
  - Unrelated HTTP links (e.g., ads, trackers, developer tools)
  - Generic footers (e.g., contact info, privacy policies, unsubscribe links)
  - Empty lines or decorative elements (e.g., "—")
- Preserve all content that is relevant and meaningful:
  - Informative or independently useful
  - Related to the topic, even tangentially
  - Provides context, background, or supporting value
  - Includes technical terms, key concepts, factual details, reasoning, and examples
- Handle mixed-relevance sentences carefully:
  - Remove only the irrelevant fragment if the rest remains coherent
  - Delete the whole sentence if the remainder loses meaning
- Do not alter meaningful content unnecessarily:
  - Only delete or modify when content is clearly meaningless or off-topic
  - Preserve the original structure, logic, and depth of the text
- Do not add explanations, notes, assumptions, or claims not found in the original text
Here is the text:
{Organic Text}
Task:
After thoroughly reading the above text, paraphrase it in high-quality and clear English following the instructions.
Start your response immediately with "Here is a paraphrased version:" and then provide the paraphrased text.

---

**DataMan Prompt**

Please score the text on fourteen evaluation criteria and specify its domain:
Text: {Text}
Domain:_
[1]Accuracy:_/5
[2]Coherence:_/5
[3]Language Consistency:_/5
[4]Semantic Density:_/5
[5]Knowledge Novelty:_/5
[6]Topic Focus:_/5
[7]Creativity:_/5
[8]Professionalism:_/5
[9]Style Consistency:_/5
[10]Grammatical Diversity:_/5
[11]Structural Standardization:_/5
[12]Originality:_/5
[13]Sensitivity:_/5
[14]Overall Score:_/5

## Structure Prompt

[Instruction]
You are given two pieces of text: an original pretraining data sample and a rephrased version. Your task is to judge if the rephrased version preserves the **structure** of the original sample.
- By "structure", we mean formatting, style, and presentation (e.g., paragraphing, JSON, list format, code blocks, markdown usage, plain text style).
- Do NOT consider semantic meaning. Ignore whether the words are the same or the content is equivalent.
- Focus only on whether the rephrased sample follows the same textual structure as the original (e.g., if the original is plain text paragraphs, the rephrased should also be plain text; if the original has bullet lists, the rephrased should also have bullet lists).
[Output]
Output **only** `1` if the structure is preserved.
Output **only** `0` if the structure is not preserved.
[Examples]
Example 1:
Original:
This is a paragraph.
This is another line.
Rephrased:
Here is a rewritten paragraph.
Here is another line of text.
Explanation: Both are plain text paragraphs, no special formatting. Structure preserved.
Output: 1
—
Example 2:
Original:
- Item one
- Item two
Rephrased:
First item. Second item.
Explanation: The original uses a bullet list, while the rephrased is plain sentences. Structure not preserved. Output: 0
—
Example 3:
Original:
{{"name": "Alice", "age": 30}}
Rephrased: {{"person": "A.", "years": 30}}
Explanation: Both are JSON objects with the same structured format. Structure preserved.
Output: 1
Explanation: The original is plain code with no markdown fences, while the rephrased introduces code fences. Structure not preserved. Output: 0
[Original]
{Organic Text}
[Rephrased]
{Recycled Text}

## Structure Classification Prompt

Consider the following web page:
Content:
{Text}
Your task is to carefully classify the structure of the given web page. Here, structure refers to the way the content is represented — for example, its markup language, formatting conventions, or encoding style — rather than its topic or purpose. The structure should NOT be HTML.
Please respond with only the name of the structure type, without any additional explanation, commentary, or extra text.

## Operation Classification Prompt

Based on the original text and its rephrased version, extract the key operations that were performed to transform the original text into the rephrased text.
Each operation should be described as one verb + one noun, e.g., "removing ads".
Focus on significant changes such as rewording, restructuring, removing, or clarifying content, while ignoring minor edits like punctuation or spacing adjustments.
Respond strictly in JSON format: {
    "operations": [
        operation1,
        operation2,
        operation3,
        ...
    ]
}
[Original Text]: {Organic Text}
[Rephrased Text]: {Recycled Text}

## WRAP Prompt

For the following paragraph give me a diverse paraphrase of the same in high quality English language as in sentences on Wikipedia.
Here is the paragraph:
{Text}
Start your response immediately with "Here is a paraphrased version:" and then provide the paraphrased text.

**ReWire Prompt**

Below is a draft from an AI Assistant when trying to accomplish task or solving a problem. Analyze and understand the task and problem(s) to be solved. Then pretend to be the expert who is most skillful to acomplish this task, write down the detailed thinking process and internal monologue that went into identifying a strategy and lay out a plan about how to solve this problem. Experts usually apply meta-reasoning and planning to reason about how to best accomplish the task before jumping to solution.

Deliberate meta-reasoning also involves reflection which can help identify issues and take a step back to explore other paths. Below are some generic examples of starting questions experts could ask themselves during meta-reasoning process. The expert will come up with the most relevant questions that can help with their thinking process, which are also very specific to the task.

Let's first try to understand the task and exactly what problem(s) to be solved. What is the core issue or problem that needs to be addressed? What are the key assumptions underlying this problem?

How can I break down this problem into smaller, more manageable parts? How can I simplify the problem so that it is easier to solve?

What kinds of solution typically are produced for this kind of problem specification? Given the problem specification and the current best solution, have a guess about other possible solutions. Let's imagine the current best solution is totally wrong, what other ways are there to think about the problem specification.

What is the best way to modify this current best solution, given what you know about these kinds of problem specification?

Am I on the right track? Let's check our progress so far.

Let's make a step by step plan and implement it with good notion and explanation.

Finally, write an improved response after thinking about how to accomplish the task. Take information and details from the original draft whenever they are useful. Therefore, the improved response should not be shorter than the original response. The improved response should have better formatting and readability, with more coherent and in-depth reasoning, while removing any noise or digression. Note that the best experts chosen to answer each prompt may be different, so please make sure the you do not sound like the same expert for all tasks.

IMPORTANT: Start your analysis and thinking right away. DO NOT add any filler text, explanations or notes about your response. Put the thinking and planning between <thinking starts> and <thinking ends>, and the improved response between <improved response starts> and <improved response ends>.

Original Draft: {Text}

**REPRO Prompt + Reward Descriptions**

Your task is to read and paraphrase the provided text following these instructions:
- Delete clearly irrelevant content:
  - Website headers, navigation bars, or menu items (e.g., "Home — About — Contact")
  - Unrelated HTTP links (e.g., ads, trackers, developer tools)
  - Generic footers (e.g., contact info, privacy policies, unsubscribe links)
  - Empty lines or decorative elements (e.g., "—")
- Preserve all content that is relevant and meaningful:
  - Informative or independently useful
  - Related to the topic, even tangentially
  - Provides context, background, or supporting value
  - Includes technical terms, key concepts, factual details, reasoning, and examples
- Handle mixed-relevance sentences carefully:
  - Remove only the irrelevant fragment if the rest remains coherent
  - Delete the whole sentence if the remainder loses meaning
- Do not alter meaningful content unnecessarily:
  - Only delete or modify when content is clearly meaningless or off-topic
  - Preserve the original structure, logic, and depth of the text
- Do not add explanations, notes, assumptions, or claims not found in the original text
————————————————

REQUIRED CRITERIA:
**1. Data Quality** When paraphrasing, keep in mind several general quality aspects. Strive for accuracy so the rewritten content reflects the intended meaning of the original. Maintain coherence so the text flows logically and remains easy to follow. Use clear, consistent, and professional language that aligns with the tone of the original. Aim for focused and precise wording without unnecessary repetition or drift. These aspects are meant to guide the paraphrasing process rather than enforce strict rules.
**2. Semantic Faithfulness** The paraphrased text should preserve all essential meaning, intent, and information from the original. It should not introduce new ideas, remove key points, or reinterpret the content.
**3. Structural Faithfulness** The paraphrased output should follow the same structure and formatting style as the original text. Paragraph breaks, list formats, and any JSON, code blocks, or markdown elements should be maintained to keep the organizational layout consistent.
**4. Length Faithfulness** The paraphrased text should remain close in length to the original and maintain a similar level of detail. Do not add new content, explanations, or elaborations, and do not significantly expand or shorten the material.
————————————————

Here is the text:
{Organic Text}
Task:
After thoroughly reading the above text, paraphrase it in high-quality and clear English following all instructions and criteria.
Start your response immediately with "Here is a paraphrased version:" and then provide the paraphrased text.

# D    CASE STUDY

## D.1    BEHAVIOR OF REPRO RECYCLING

In this section, we present more representative examples showing how REPRO transforms organic data into higher-quality rephrasings. The quality percentile reflects how an example ranks relative to others in the pool based on its fastText score, with a higher percentile indicating better quality. For low-quality organic data, we observe that REPRO can clarify unclear information or logical gaps. For moderate-to-low quality data that often contains irrelevant webpage artifacts (such as navigation text or image placeholders), REPRO removes these elements during rephrasing while keeping the essence of the text. For already high-quality data, REPRO reorganizes and paraphrases the content to improve clarity and further raise the quality score. Across these cases, as well as many more examined through manual inspection, we did not observe issues such as degeneration, repeated formatting patterns, or reduced diversity. Through RL optimization, REPRO preserves the key characteristics of organic data (writing style, structure, genre, etc.) and learns to flexibly apply various operations to polish the text.

---

**REPRO: Low Quality → Moderate Quality**
Operations: *Remove, Paraphrase*

| **Organic: Quality Percentile=16.3%** | **Recycled: Quality Percentile=33.2%** |
|---|---|
| Kate Upton 'Bombshell Beauty' Represents Us Elle September 2013! (Pics) Image00001American swim suit model and actress Kate Upton represents Elle September for 2013.    She has been photographed by Carter Smith and styled by Joe Zee.Image00001Image00001Image00001 Speak Your Mind | Kate Upton, the renowned American swimwear model and actress, graces the cover of Elle magazine for September 2013. Photographed by Carter Smith, she is styled by Joe Zee, embodying the essence of "Bombshell Beauty" in the issue. |

---

**REPRO: Low Quality → High Quality**
Operations: *Remove, Clarify*

| **Organic: Quality Percentile=4.8%** | **Recycled: Quality Percentile=97.9%** |
|---|---|
| Interview Question
Interview Poway, CA
Shift 0xFF to the right by 2, what do you get?
Interview Answer
1 Answer
i THINK since FF is 1111 1111, and you get 0011 1111, you end up with 0x3F
Interview Candidate on Nov 10, 2011
Add Answers or Comments
To comment on this, Sign In or Sign Up. | Interview Question
Interview Location: Poway, CA
Question: Shift the value 0xFF to the right by 2 bits. What is the result?
Answer:
Consider the binary representation of 0xFF, which is 1111 1111. When you shift this binary number to the right by 2 bits, the result is 0011 1111. This binary number corresponds to the hexadecimal value 0x3F.
Interview Candidate: November 10, 2011 |

**REPRO: Moderate Quality → High Quality**
Operations: *Remove, Paraphrase*

**Organic: Quality Percentile=35.8%**

Observation and Growth in Scientific Knowledge

In the writings of scientists we find claim to the effect that we can observe items such as pulsars, gravity waves, quarks, electrons, etc. An epistemological theory, originally developed by Dretske and modified by Jackson, is used to give an account of such claims and the extent to which they may be deemed correct. The theory eschews talk of the theory-ladenness of observation while giving an account of how our observation reports may evolve with growth in scientific knowledge. The theory is partially linked to Shapere's recent account of observation in science.

Keywords No keywords specified (fix it)
Categories (categorize this paper)
DOI 10.2307/193125
Save to my reading list
Follow the author(s)
My bibliography
Export citation
Find it on Scholar
Edit this record
Mark as duplicate
Revision history Request removal from index
Download options

**Recycled: Quality Percentile=95.7%**

Observation and Growth in Scientific Knowledge

In scientific literature, we often encounter assertions about the ability to observe phenomena like pulsars, gravity waves, quarks, and electrons. This discussion explores an epistemological theory originally proposed by Dretske and later refined by Jackson, which is employed to analyze such claims and evaluate their validity. The theory avoids the notion that observation is inherently theory-laden, instead explaining how observation reports can evolve alongside the expansion of scientific knowledge. It also partially connects to Shapere's recent approach to understanding observation in scientific contexts.

---

**REPRO: High Quality → Higher Quality**
Operations: *Reorganize, Paraphrase*

| **Organic: Quality Percentile=83.2%** | **Recycled: Quality Percentile=95.1%** |
|---|---|
| Preparation Instructions | Preparation Instructions |
| Preheat oven to 325°F. Heat 2 Tablespoons peanut oil in a large (5 quart+) oven proof Dutch oven over medium heat. Lightly brown meat in batches. Do not over crowd or meat will just release juices and not brown. Transfer browned meat to a large bowl. Put remaining 1Tablespoon of peanut oil into the pot and sauté the onions, garlic and ginger until the onions are clear. Meanwhile, thoroughly whisk together the Cake Spice, ketchup, sherry, tamari or soy sauce, sesame oil and honey. Taste, may taste a bit salty but the red potatoes will absorb this salt. Adjust for honey. Put onion mixture and potatoes into the large bowl. Pour the whisked sauce over the stew and stir or toss to coat. Pour stew back into the Dutch oven and bake in the oven for 3 hours or until the meat and vegetables are tender. To serve garnish with sliced green onions. | 1. Preheat your oven to 325°F. In a large ovenproof Dutch oven (5 quarts or larger), heat 2 tablespoons of peanut oil over medium heat. Brown the meat in batches without overcrowding, as this will prevent the juices from releasing and ensure proper browning. Transfer the browned meat to a large bowl. 2. In the same pot, add the remaining 1 tablespoon of peanut oil and sauté the onions, garlic, and ginger until the onions become transparent. 3. Meanwhile, whisk together the Cake Spice, ketchup, sherry, tamari, or soy sauce, sesame oil, and honey. The mixture may taste slightly salty, but the red potatoes will absorb the excess. Adjust the seasoning with honey to your preference. 4. Combine the sautéed onion mixture with the potatoes in the large bowl and pour the whisked sauce over the stew, stirring or tossing to ensure everything is well-coated. 5. Transfer the stew back into the Dutch oven and bake for 3 hours, or until the meat and vegetables are tender. Serve garnished with sliced green onions. |

D.2  BEHAVIOR OF DIFFERENT REPHRASING METHODS

Finally, we study the behavior of different rephrasing methods. The recycled text from ReWire or the rephraser trained without faithfulness rewards introduces extraneous information loosely related to the original text (e.g., the assassination of Archduke Franz Ferdinand) and alters the structure of the organic data from markdown to plain text. This may increase the quality score but reduce the overall faithfulness of recycled data, posing a risk of model collapse. WRAP also adds content that is not present in the organic data. In contrast, REPRO strictly adheres to the original content and structure while improving the writing quality by removing irrelevant advertisements and metadata.

- Gallipoli is a peninsular in Turkey
- British and colonial forces attacked the peninsula in 1915 during World War I
- It was known as the Dardanelles Campaign
- Deaths were 43,000 British, 15,000 French, 8,700 Australians, 2,700 New Zealanders and 1,370 Indians
- The Allies withdrew from the campaign which is celebrated on ANZAC Day
blog comments powered by Disqus","url":"http://5factsabout.com/articles/5-facts-about-Gallipoli/","warcinfo":"robots: classic
hostname: ip-10-137-6-227.ec2.internal
software: Nutch 1.6 (CC)/CC WarcExport 1.0
isPartOf: CC-MAIN-2015-40
operator: CommonCrawl Admin
description: Wide crawl of the web for September 2015
publisher: CommonCrawl
format: WARC File Format 1.0
conformsTo: http://bibnum.bnf.fr/WARC/WARC_ISO_28500_version1_latestdraft.pdf

### Recycled Data (REPRO)

**Gallipoli: Key Historical Facts**
Gallipoli is a peninsula in Turkey that played a significant role in military history.
During World War I, British and colonial forces launched an attack on the region in 1915 as part of the Dardanelles Campaign.
The campaign resulted in substantial casualties, including 43,000 British, 15,000 French, 8,700 Australian, 2,700 New Zealand, and 1,370 Indian soldiers.
Despite initial advances, the Allies ultimately withdrew from the campaign.
The event is commemorated annually on ANZAC Day.

### Recycled Data (REPRO w/ SFT)

5 Key Points About Gallipoli

1. **Location**: Gallipoli is a peninsula situated in Turkey.
2. **Historical Conflict**: It was the site of an attack by British and colonial forces during World War I in 1915, known as the Dardanelles Campaign.
3. **Casualties**: The campaign resulted in significant losses, with 43,000 British troops, 15,000 French soldiers, 8,700 Australians, 2,700 New Zealanders, and 1,370 Indian soldiers killed.
4. **Military Withdrawal**: Due to the heavy casualties and strategic challenges, the Allies decided to withdraw from the campaign.
5. **Memorial Day**: This historical event is commemorated annually on ANZAC Day.

**Recycled Data (Prompting)**

5 Facts About Gallipoli

The following are five facts regarding Gallipoli:

- Gallipoli is a peninsula located in Turkey.
- British and colonial troops launched an attack on the peninsula in 1915 during World War I.
- The campaign is referred to as the Dardanelles Campaign.
- The total number of casualties included 43,000 British, 15,000 French, 8,700 Australians, 2,700 New Zealanders, and 1,370 Indians.
- The Allied forces eventually withdrew from the campaign, which is commemorated on ANZAC Day.

**Recycled Data (Prompting w/ Thinking)**

<think>
Okay, the user wants me to paraphrase the given text following specific instructions. Let me start by understanding the original text.

The original text has a heading "Related Ads" and then "5 Facts About Gallipoli". The main content lists five facts about Gallipoli. There's also a mention of blog comments powered by Disqus, which is a footer or unrelated link.

First, I need to delete irrelevant content. The "Related Ads" header is probably not needed. The "blog comments powered by Disqus" is a footer, so I should remove that. Also, check for any empty lines or decorative elements. The text seems to have a bullet point list, which is relevant.

Now, the task is to preserve all meaningful content. The five facts are important. Let me check each point:

1. Gallipoli is a peninsular in Turkey – that's a fact.
2. British and colonial forces attacked in 1915 during WWI – relevant.
3. Known as Dardanelles Campaign – correct.
4. Deaths listed – important details.
5. Allies withdrew and it's celebrated on ANZAC Day – key info.

I need to paraphrase these points without changing the meaning. Also, ensure that the structure is maintained but in a different way. For example, instead of using "Here are 5 facts", maybe start with "Gallipoli, a peninsula in Turkey, was the site of a significant military campaign during World War I." Then list the facts concisely.

Check for any mixed-relevance sentences. The original text has the "blog comments" part, which is irrelevant, so remove that. The rest is all relevant.

Make sure not to add any explanations or assumptions. Just rephrase the existing content. Also, preserve technical terms like "Dardanelles Campaign" and the numbers.

So the final paraphrase should start with the header "Here is a paraphrased version:" followed by the rephrased facts. Let me put that together now.
</think>
Here is a paraphrased version:
Gallipoli, a peninsula in Turkey, was the site of a significant military campaign during World War I. British and colonial forces launched an attack in 1915, known as the Dardanelles Campaign. The conflict resulted in approximately 43,000 British, 15,000 French, 8,700 Australian, 2,700 New Zealand, and 1,370 Indian casualties. The Allied forces eventually withdrew from the campaign, which is commemorated on ANZAC Day.

**Recycled Data (RL w/o Faithfulness)**

Gallipoli is a strategically significant peninsula located in northwest Turkey, situated between the Aegean Sea and the Dardanelles Strait. This location played a pivotal role during the First World War, specifically during the 1915 campaign known as the Dardanelles or the Battle of Gallipoli, a major Allied military operation initiated by Britain and its colonial allies. The objective was to open a new front in the Ottoman Empire to relieve pressure on the Russian Empire and to liberate Constantinople from Turkish control.

The Gallipoli campaign saw a significant deployment of British, French, Australian, New Zealand, and Indian troops. The operation commenced on May 19, 1915, with the aim of capturing the Turkish-held Gallipoli Peninsula and forcing the Ottomans to withdraw from the Eastern Front. However, the campaign faced numerous challenges, including treacherous terrain, strong Turkish defenses, and logistical difficulties, which contributed to a costly stalemate.

The Allied forces suffered heavy casualties during the campaign, with estimates ranging from 43,000 British, 15,000 French, 8,700 Australians, 2,700 New Zealanders, and 1,370 Indian soldiers killed, in addition to numerous wounded. Despite their efforts, the Allies were unable to achieve their strategic objectives, and the campaign ended in disaster, leading to the withdrawal of British and French forces from the peninsula in January 1916.

This pivotal event in military history has been immortalized in Australian and New Zealand military traditions, and ANZAC Day, observed on April 25th, commemorates the service and sacrifice of these nations' soldiers in the First World War, including the gallantry displayed at Gallipoli. The commemorations emphasize themes of bravery, sacrifice, and national identity, serving as a reminder of the significant contributions made by Australian and New Zealand soldiers to the Allied war efforts.

**Recycled Data (fastText as Quality Reward)**

The text outlines five significant facts regarding Gallipoli. First, it describes Gallipoli as a geographical region located in Turkey. Additionally, it highlights the historical context of the area, specifically the military operations that took place there during World War I. Furthermore, the text provides a detailed account of the casualties sustained during the conflict, emphasizing the substantial loss of life on both sides. Finally, it mentions the historical significance of the campaign, including the eventual withdrawal of the Allied forces and its connection to ANZAC Day. Overall, the text serves as a concise summary of the key points related to Gallipoli's historical importance.

**Recycled Data (WRAP)**

Five Noteworthy Factoids Concerning Gallipoli

Gallipoli, a coastal region in Turkey, has been the focus of five intriguing facts that have garnered significant attention.

Firstly, Gallipoli is a peninsula located in Turkey, and it has become the focal point of historical events, particularly during World War I.

Secondly, the British and colonial forces launched an attack on the peninsula in 1915, which was known as the Dardanelles Campaign.

Thirdly, the casualties during the campaign were significant, with 43,000 British, 15,000 French, 8,700 Australians, 2,700 New Zealanders, and 1,370 Indians losing their lives.

Fourthly, the Allies withdrew from the campaign, marking a significant moment in the history of World War I.

Finally, the campaign has become a symbol of bravery and sacrifice, and it is celebrated annually on ANZAC Day, which stands for Australian and New Zealand Army Corps Day.

The Gallipoli campaign was a pivotal event in World War I, marking a significant turning point in the conflict. Located on the Gallipoli peninsula in Turkey, the campaign was fought between the Allied Powers, including British, French, Australian, New Zealand, and Indian forces, and the Ottoman Empire.

To understand the context of the campaign, it is essential to consider the events leading up to the war. In 1914, the assassination of Archduke Franz Ferdinand sparked a chain reaction of diplomatic crises and military mobilizations, eventually leading to the outbreak of World War I. The Allied Powers, seeking to gain a strategic advantage, set their sights on the Dardanelles, a narrow strait that connected the Aegean Sea to the Sea of Marmara, and ultimately, to the Black Sea.

The Gallipoli campaign, also known as the Dardanelles Campaign, began on February 19, 1915, when British and French warships launched a bombardment of the Ottoman defenses. The campaign was intended to capture the strategic location of Constantinople, now Istanbul, and secure a vital trade route. However, the Ottoman forces, led by Mustafa Kemal, were better prepared than anticipated, and the campaign quickly turned into a disastrous defeat for the Allies.

The campaign itself was marked by a series of bloody battles, including the landing at Anzac Cove, the Battle of Lone Pine, and the Battle of Chunuk Bair. Despite being outnumbered, the Ottoman forces held their ground, and the Allies suffered heavy casualties. The death toll was staggering, with 43,000 British, 15,000 French, 8,700 Australian, 2,700 New Zealand, and 1,370 Indian soldiers losing their lives.

The aftermath of the campaign was marked by the withdrawal of the Allied forces in January 1916. The failure of the campaign had significant consequences, including the resignation of Winston Churchill, who had been a strong advocate for the campaign. The legacy of the Gallipoli campaign is still commemorated today, with ANZAC Day, April 25, marking the anniversary of the landing at Anzac Cove.

In conclusion, the Gallipoli campaign was a pivotal event in World War I, marking a significant turning point in the conflict. The campaign's failure had far-reaching consequences, including the resignation of key leaders and a shift in the balance of power. Today, the campaign is remembered as a testament to the bravery and sacrifice of the soldiers who fought and died on the Gallipoli peninsula.

