# OpenReview forum: "RePro: Training Language Models to Faithfully Recycle the Web for Pretraining"
_ICLR.cc/2026/Conference — Submitted to ICLR 2026_

### Official Review · Reviewer_J499 · 2025-10-26

**Soundness:** 3
**Presentation:** 4
**Contribution:** 2
**Rating:** 6
**Confidence:** 4

**Summary:**

This paper discusses the problem of finite internet data to pretrain LLMs on. It motivates the expansion of current finite data in a way that is efficient (unlike GPT4/5 rephrasing everything) and faithful i.e. preserve underlying semantics. Towards these, they train a Qwen3/4B to rephrase data that and carry out this data augmentation to pretrain models. They use RL based loss that uses
two metrics viz. DataMan [1] score for quality assessment and BERTScore [2] for semantic similarity check. The paper name their method as "RePRO".

For experiments, they pretrain a 400M and 1.4B model and compare against baseline method (viz ReWire, which prompts llama3/70B) and baseline dataset which is standard pretraining corpus (?) and report better benchmark numbers.



[1] Ru Peng, Kexin Yang, Yawen Zeng, Junyang Lin, Dayiheng Liu, and Junbo Zhao. DataMan: Data
manager for pre-training large language models. In Proc. of ICLR, 2025
[2] Tianyi Zhang, Varsha Kishore, Felix Wu, Kilian Q Weinberger, and Yoav Artzi. BERTScore: Eval-
uating text generation with bert. In Proc. of ICLR, 2020

**Strengths:**

- Very well written paper, have cited important related works appropriately, even outside related work section to motivate and compare thoughtfully.
- Nice set of ablation, easy to verify where the gains comes from
- Gain on performance (albeit marginal) against proper baselines. I also appreciate the case study in Appendix D.

All in all, this paper proposes how RLing a rephrases can make them better at augmenting internet scrap like data while optimizing for quality and faithfulness, among others.

**Weaknesses:**

See below

**Questions:**

I like this paper, to better understand if the gains are actually from the proposed method, I would need response to following questions:

- Is the Table 3 "Prompting" is prompting unoptimized Qwen3/4B with same prompt (as used for RL training)? What sampler did you use? Is it 0-temperature sampling? Can I ask to see performance with temperature 0.6, and also another with top-p 0.96 to clearly understand the gains are from RL and not just sampling. Even if this is the case, this should be weakness of RL methods in general applicable here and not of RePro directly.
- Figure 3 is missing bars for WRAP, ProX, and ReWire (+ Prompting baseline), it would be nice to see how RePro comapres to these across scales.
- Is the thinking mode of qwen3 used for this? Can I see the thinking traces for the case study in Appendix D?

---

> ### Author Response · Authors · 2025-11-22
> **Official Rebuttal to Reviewer J499**
>
> Thank you for your time and insightful review of our paper! We address your questions/comments below:
>
> > **Question 1-1:** Is "Prompting" in Table 3 unoptimized Qwen3-4B with the same prompt used for RL?
>
> **Response:** Yes. The prompting baseline refers to unoptimized Qwen3-4B with the exact same prompt used for RL. To make the comparison with RL more informative (suggested by Reviewer `A8Ei`), we strengthen this baseline by adding the descriptions of our reward functions directly into the prompt (see the full prompt at the end of Appendix C.4). The results show that this enhanced prompting baseline (denoted by Prompting+) improves Core accuracy less than 0.1% compared to our base prompt, and thus RePro still significantly outperforms this enhanced baseline. Our results indicate that the unoptimized model may not be able to reliably satisfy all the criteria through prompting alone, so a dedicated RL procedure is required to shape it into a faithful and high-performing rephraser.
>
> | **400M Setting** | **Commonsense Reasoning** | **Language Understanding** | **Reading Comprehension** | **Symbolic Problem** | **World Knowledge** | **Core**    |
> | ---------------- | ------------------------- | -------------------------- | ------------------------- | -------------------- | ------------------- | ----------- |
> | Organic   | 0.23613       | 0.27079 | 0.03724       | 0.14535  | 0.20126 | 0.18990     |
> | Prompting | 0.24310       | 0.26758 | 0.05075       | 0.17392  | 0.20196 | 0.19847     |
> | Prompting+       | 0.27831       | 0.25229 | 0.06870       | 0.16475  | 0.20032 | 0.19910     |
> | RePro     | **0.28454**   | **0.27792**    | **0.07181**   | **0.19409**   | **0.21154**  | **0.21658** |
>
> > **Question 1-2:** Performance with temperature 0.6, and also another with top-p 0.96 to clearly understand the gains are from RL and not just sampling.
>
> **Response:** Thank you for your suggestion. Our original configuration uses temperature 1.0 and top-p 0.9, following the settings in ReWire. We have added these hyperparameters in Section 4. To better understand the effect of sampling, we vary the temperature to 0.6 and the top-p value to 0.96 as per your request. A temperature of 0.6 produces more deterministic and consistent outputs, while top-p 0.96 leads to more diverse and creative sampling. As shown in the results below, top-p=0.96 performs slightly better than the other two, likely due to the increased diversity in the recycled data, although the overall differences are small. RePro remains clearly stronger than all prompting-based variants, showing that RL is a more practical path for training an effective rephraser. We will add this ablation in the Appendix.
>
> | **400M Setting**    | **Commonsense Reasoning** | **Language Understanding** | **Reading Comprehension** | **Symbolic Problem** | **World Knowledge** | **Core** |
> | ------------------- | ------------------------- | -------------------------- | ------------------------- | -------------------- | ---| ----- |
> | temp=1.0 top-p=0.9  | 0.24310       | 0.26758 | 0.05075       | **0.17392**  | **0.20196** | 0.19847  |
> | temp=0.6 top-p=0.9  | **0.26988**   | 0.25236 | **0.07081**       | 0.16191  | 0.19956 | 0.19743  |
> | temp=1.0 top-p=0.96 | 0.26594       | **0.27233** | 0.04183       | 0.16991  | 0.20038 | **0.20040**  |
>
> > **Question 2:** Figure 3 is missing bars for WRAP, ProX, ReWire, and Prompting baselines
>
> **Response:** Thank you for your suggestion. We have added these baselines (except ReWire) to Figure 3. The key takeaway is that RePro achieves the largest improvement over baseline methods when the amount of unique tokens is 21.6B (7.2B high-quality organic + 14.4B recycled). This demonstrates that RePro can generate more high quality tokens from a fixed organic pool than previous web recycling methods, indicating better data efficiency. The gap becomes slightly smaller in the 28.8B setup, likely because all methods may include some moderate-to-low quality data at this data scale, which reduces the relative advantage of higher quality recycled tokens.
>
> We do not include ReWire in this comparison, since they only release the top-7.2B selected recycled data $\mathcal{D}\_{\text{rec-hq}}$, but not the full recycled pool $\mathcal{D}\_{\text{rec}}$. Therefore, we cannot select the top-14.4B or top-21.6B tokens required in Figure 3. Additionally, fully reproducing their rephrasing pipeline is beyond our available compute budget, which requires more than 60k GPU hours as shown in Table 4. We are willing to include its comparison in Figure 3 if they release the full recycled pool.
>
> | **Unique Token** | **14.4B (7.2B + 7.2B)** | **21.6B (7.2B + 14.4B)** | **28.8B (7.2B + 21.6B)** |
> | -- | --------- | --------- | --------- |
> | Organic | 0.27108   | 0.28049   | 0.26739   |
> | Prompting | 0.28805   | 0.28250   | 0.27811   |
> | WRAP | 0.28335   | 0.27910   | 0.27515   |
> | ProX | 0.29004   | 0.28269   | 0.27790   |
> | RePro | **0.29929**   | **0.29743**   | **0.28346**  |

---

> ### Author Response · Authors · 2025-11-22
> **Official Rebuttal to Reviewer J499 (Continued)**
>
> > **Question 3:** Are the thinking modes of Qwen3 used?
>
> **Response:** No. We add `no_think` at the end of our system prompt to disable the thinking mode, as we have not identified a clear advantage of using it in our preliminary experiments. For direct prompting, we find that thinking mode often alters the structure of the organic text (see page 27 in our updated PDF), which is unpreferred. For RL, we didn’t see a clear performance gain of the rewards when enabling the thinking mode. This may be because the RL objective is shaped entirely by the final text-level rewards, and the model can reliably explore and adjust its rephrasing behavior based on output feedback alone. Furthermore, enabling thinking mode may introduce additional inference costs for our rephrasing. Thank you for your observation.

---

### Official Review · Reviewer_A8Ei · 2025-10-30

**Soundness:** 2
**Presentation:** 3
**Contribution:** 2
**Rating:** 4
**Confidence:** 4

**Summary:**

This paper focuses on generating new pretraining data by rephrasing existing data. Specifically, the authors train a rephraser LM (4B) using RL to generate effective and faithful rephrasing of existing pretriaining data. Four reward signals, one focusing on quality and the other three focusing on faithfulness are used for RL training. Using this model, authors rephrase 72B tokens sampled from DCLM-RefinedWeb, and show that models trained with high quality rephrased data perform better than the models trained with the original data. Specifically, proposed rephrasing leads to 2-3x organic data efficiency, and also outperfoms ReWire, an existing approach that uses 70B model as rephraser.

**Strengths:**

Proposes an interesting approach to train a rephraser model and demonstrates the effectiveness of the trained model by comparing it with alternative rephrasing strategies

Conducts ablation studies demonstrating the effectiveness of the reward signals used for RL

**Weaknesses:**

* According to Sec 3.1, the entire organic pool is recycled including the high quality data D_{org-hq} (Eq. 2). The rephrased data subset used for training is selected as the highest quality samples from the entire rephrased pool (eq. 3). With this strategy, D_{rec-hq} could be dominated by rephrased version of D_{org-hq} rather than the rephrased version of D_org - D_{org-hq}. This seems to go against the goal of taking advantage of low-quality samples by rephrasing them. Authors should show how many tokens in D_{rec-hq} are derived from D_{org-hq} and how many are derived from D_org - D_{org-hq}.
    * Is the proposed approach actually leveraging additional low-quality (but diverse) data or mainly using rephrased version of already high quality samples?
    * How would the results look if D_{rec-hq} was selected only from  D_org - D_{org-hq}?


* **Do we really need RL here or do we just need to clearly communicate with the rephraser?** - The proposed rephraser is trained with RL using four reward functions. For a fair comparison, both prompting-based rephrasing baseline and GPT prompting-based SFT data generation should incorporate the knowledge of the rewards into the prompt used for rephrasing. This could be done by describing what the reward functions are trying to capture as part of the prompt (which the authors are already doing for prompting the reward computing LLMs). This will give a clear signal if RL training is really needed or we simply need to tell the rephraser clearly what it needs to optimize when rephrasing by describing the rewards in the prompt.


* **Relation between RL reward function and the final data quality function used for selecting sample** - According to experimental results, directly using DataMan for selecting samples is worse than using DCLM-fastText. However, training with DCLM-fastText is worse than training with DataMan. This suggests that there is some disagreement between the two which warrants further attention/analysis. Authors could look at the correlation (or scatter plot) between the two to get a better understanding of this behavior

* Experiments are conducted using small models (400M and 1B), while the rephraser itself is a much bigger 4B model. It is unclear if the proposed approach would be effective for rephrasing datasets when training bigger models (let’s say 7B).


Minor suggestion:
The comparison of pretrainig data with fossil fuel is not meaningful. Used fossil fuel disappers from earth and hence the reserves are going down. In contrast, pretraining data does not disappear after being used for training.

**Questions:**

Is the proposed approach actually leveraging additional low-quality (but diverse) data by rephrasing?

Do we really need RL here or do we just need to clearly communicate with the rephraser?

Are DataMan and DCLM-fastText scores correlated or not?

---

> ### Author Response · Authors · 2025-11-22
> **Official Rebuttal to Reviewer A8Ei**
>
> Thank you for your time and insightful review of our paper! We address your questions/comments below:
>
> > **Weakness 1 & Question 1:** Is RePro actually leveraging additional low-quality (but diverse) data by rephrasing?
>
> **Response:** Thank you for your notice on the recycling of low-quality data. We provide a comparison of the scatter plots showing the relationship between the fastText scores before and after rephrasing with each method, and we measure how much of the high-quality recycled data $\mathcal{D}\_{\text{rec-hq}}$ comes from non-high-quality organic data $\mathcal{D}\_{\text{org}} - \mathcal{D}\_{\text{org-hq}}$. As shown in Figure 4 (see our updated PDF), we find that for RePro, almost **two-thirds (63.5%)** of $\mathcal{D}\_{\text{rec-hq}}$ originates from $\mathcal{D}\_{\text{org}} - \mathcal{D}\_{\text{org-hq}}$. In comparison, the corresponding proportions for Prompting, WRAP, and ProX are 31.0%, 42.5%, and 39.0%, respectively, indicating that RePro converts substantially more low-quality data into high-quality rephrasings. Furthermore, we observe that RePro adopts a higher cutoff threshold for high-quality data than other methods, which suggests that our rephraser achieves a larger improvement in the overall fastText scores. This further explains why RePro demonstrates better organic data efficiency in Figure 3, as the overall quality of its rephrased data is higher than that of the other methods.
>
> We do not include ReWire in this comparison, since they only release the high-quality recycled data $\mathcal{D}\_{\text{rec-hq}}$, but not the full recycled pool $\mathcal{D}\_{\text{rec}}$. Therefore, it is difficult for us to estimate their high-quality threshold, and fully reproducing their rephrasing pipeline is beyond our available compute budget (more than 60k GPU hours).
>
> We also run new experiments on our 400M and 1B setups, where we only use $\mathcal{D}\_{\text{rec-hq}}$ that are from $\mathcal{D}\_{\text{org}} - \mathcal{D}\_{\text{org-hq}}$. We can see that the majority of our gains (**77.2%** in 400M setup and **82.4%** in 1B setup) actually comes from recycling low-quality data. Even in this case, our method can still outperform all baselines that use all their $\mathcal{D}\_{\text{rec-hq}}$. However, we would like to highlight that in a realistic setting, it is more reasonable to utilize all $\mathcal{D}\_{\text{rec-hq}}$ for a better performance, as high quality data can still be transformed for better learnability.
>
> | **400M Setting**                                             | **Commonsense Reasoning** | **Language Understanding** | **Reading Comprehension** | **Symbolic Problem** | **World Knowledge** | **Core**    |
> | ------------------------------------------------------------ | ------------------------- | -------------------------- | ------------------------- | -------------------- | ------------------- | ----------- |
> | Organic   | 0.23613  | 0.27079  | 0.03724  | 0.14535              | 0.20126  | 0.18990     |
> | RePro (use $\mathcal{D}\_{\text{rec}}$ from $\mathcal{D}\_{\text{org}} - \mathcal{D}\_{\text{org-hq}}$) | **0.28674** | 0.27051  | 0.06737 | 0.17920 | 0.20992 | 0.21050     |
> | RePro (use all $\mathcal{D}\_{\text{rec}}$) | 0.28454 | **0.27792** | **0.07181** | **0.19409**  | **0.21154** | **0.21658** |
>
> | **1B Setting**                                               | **Commonsense Reasoning** | **Language Understanding** | **Reading Comprehension** | **Symbolic Problem** | **World Knowledge** | **Core**    |
> | ------------------------------------------------------------ | ------------------------- | -------------------------- | ------------------------- | -------------------- | ------------------- | ----------- |
> | Organic  | 0.32348 | 0.38371                    | 0.19584 | 0.19795              | 0.28745  | 0.28578     |
> | RePro (use $\mathcal{D}\_{\text{rec}}$ from $\mathcal{D}\_{\text{org}} - \mathcal{D}\_{\text{org-hq}}$) | 0.35527 | 0.38393  | **0.21452** | **0.21032** | 0.29348 | 0.29691     |
> | RePro (use all $\mathcal{D}\_{\text{rec}}$) | **0.36776**               | **0.38519** | 0.20832                   | 0.20597              | **0.30304** | **0.29929** |
>
> To better understand how RePro operates on different data quality, we provide more recycling cases in our Appendix D.2. Findings include:
>
> - For low-quality organic data, we observe that RePro can clarify unclear information or logical gaps.
>
> - For moderate-to-low quality data that often contains irrelevant webpage artifacts (such as navigation text or image placeholders), RePro removes these elements during rephrasing while keeping the essence of the text.
>
> - For already high-quality data, RePro reorganizes and paraphrases the content to improve clarity and further raise the quality score.
>
> In summary, RePro filters out noisy or irrelevant content, clarifies unclear information, and refines the organization and writing of already high-quality texts to further enhance their learnability.

---

> ### Author Response · Authors · 2025-11-22
> **Official Rebuttal to Reviewer A8Ei (Continued)**
>
> > **Weakness 2 & Question 2:** Do we really need RL or do we just need to clearly communicate with the rephraser?
>
> **Response:** Thank you for your suggestion about incorporating reward descriptions into the prompting baseline. We add this comparison (denoted as Prompting+) as well as an SFT baseline trained on GPT-generated data using the same enhanced prompt (denoted as SFT+). The full prompt is provided at the end of Appendix C.4. The results show that Prompting+ yields only a negligible improvement over our base prompt, and SFT+, although much better than SFT, still remains clearly below RePro.
>
> | **400M Setting** | **Commonsense Reasoning** | **Language Understanding** | **Reading Comprehension** | **Symbolic Problem** | **World Knowledge** | **Core**    | **Avg. DataMan** | **Avg. BERTScore** |
> |  -- |  -- |  -- |  -- |  -- |  -- | -- |  -- |  -- |
> | Organic  | 0.23613  | 0.27079   | 0.03724  | 0.14535  | 0.20126 | 0.18990 | 3.976·   | - |
> | Prompting    | 0.24310  | 0.26758   | 0.05075  | 0.17392  | 0.20196 | 0.19847 | 4.072    | 0.691  |
> | Prompting+   | 0.27831  | 0.25229   | 0.06870  | 0.16475  | 0.20032 | 0.19910 | 3.988    | 0.737  |
> | SFT  | 0.24447  | 0.24920   | 0.04013  | 0.16564  | 0.21009 | 0.19216 | 4.315    | 0.559  |
> | SFT+ | 0.26049  | 0.27111   | 0.06593  | 0.16899  | 0.20206 | 0.20278 | 4.174    | 0.703  |
> | RePro    | **0.28454**   | **0.27792**    | **0.07181**   | **0.19409**  | **0.21154** | **0.21658** | **4.583**    | **0.752**  |
>
> If we look at the average quality (DataMan) and faithfulness (BERTScore) scores for prompting and SFT methods (also prompting a stronger model), we observe a consistent trade-off. For example, adding reward descriptions into the prompt can increase the average BERTScore, but it often lowers the DataMan score, which limits the rephraser’s ability to improve performance across all tasks. In contrast, our RL approach can jointly optimize both quality and faithfulness through active exploration of high-quality and faithful rephrasings. Overall, these results demonstrate that a dedicated RL procedure is more effective than prompting alone for shaping the base model into a faithful and high-performing rephraser.
>
> > **Weakness 3 & Question 3:** Are DataMan and DCLM-fastText scores correlated or not?
>
> **Response:** Thank you for your notice on the correlation between quality metrics. First, we would like to provide more details about DCLM-fastText. DCLM-fastText is a n-gram-based classifier trained on DCLM-RefinedWeb (used as negative examples) and OpenHermes 2.5 instruction-following data + ExplainLikeImFive QA data (used as positive examples). Consequently, it is optimized specifically for mining high-quality data within DCLM-RefinedWeb. This explains why it performs well as a selector on DCLM-RefinedWeb, but it also means that its generalization ability, especially when used as a reward in RL, is limited. For instance, it tends to assign high scores to any QA-style or instruction-response text regardless of its actual content quality. In an RL setup, this behavior can lead to reward hacking issues and reduced diversity in rephrasing, since not all web text should be converted into QA or instruction formats. In contrast, our quality reward, DataMan, is a more robust and holistic metric that evaluates text quality using a language model across 13 criteria and is less likely to overfit to any particular corpus, making it better suited for RL training. Despite this difference, our recycled data has a significant improvement in overall fastText scores (see our response to Question 1), indicating that high-quality rephrasing generalizes well across different quality metrics.
>
> To further illustrate the differences between these two metrics, we compute the average fastText score within each DataMan score bucket using 30,000 samples randomly drawn from the organic data pool. The correlation between the two metrics is indeed not very high (Spearman=0.248), indicating that they capture different aspects of quality. Specifically, for DataMan = 1, the average fastText score is the lowest, and for DataMan = 5, it is the highest, showing that the two metrics agree at some edge cases. However, for the middle buckets (DataMan = 2, 3, and 4), the corresponding average fastText scores do not align well with the DataMan values. This discrepancy supports our argument that the two scorers assess quality from different perspectives: DCLM-fastText focuses on identifying formats or surface phrases associated with high-quality data, while DataMan focuses more on semantic value and overall textual informativeness. In edge cases, these two views can coincide, but for more ambiguous middle cases they often diverge, leading to the observed mismatch. We will add this discussion in the paper.
>
> | DataMan Bucket   | 1  | 2  | 3 | 4  | 5  |
> |  -- | -- | ---------- | --------- | ---------- | ---------- |
> | Avg. fastText Score | 0.01628565 | 0.02057239 | 0.0195243 | 0.01637771 | 0.02445352 |

---

> ### Author Response · Authors · 2025-11-22
> **Official Rebuttal to Reviewer A8Ei (Continued)**
>
> > **Weakness 4:** RePro for training bigger models
>
> **Response:** Thank you for your suggestion. To address this question, we follow the DCLM 3B-1x setup (2.8B model, 55.9B tokens) and report the results below (also in the updated PDF). RePro still holds the leading position among all web recycling methods, doubling the gains of the second-best baseline (ProX). We also observe that in this larger setup, Wikipedia-style rephrasing (WRAP) can even harm performance, likely due to its limited diversity and weaker preservation of the original distribution, both of which become increasingly important for larger pretraining models.
>
> We would like to note that, compared to the 400M setup, the 3B setup is **undertrained** due to our compute constraints, which may limit the absolute performance gains across all methods. Therefore, we also follow the optimal organic-to-rephrased mixture ratio (2:1) identified by prior work [1] to better reveal the potential of our method. Under this configuration, RePro achieves a **4.7%** relative gain over the organic-only baseline. In summary, RePro consistently outperforms existing baselines in a larger pretraining setup and shows strong scalability and generalization ability of a relatively small but well-optimized rephraser. Unfortunately, during the short rebuttal phase, we don’t have enough compute resources to further train more tokens on 3B setup or scale our method up to DCLM 7B-1x, but we are willing to incorporate larger-scale results if we get more resources.
>
> | **3B Setting** | **Pool** | **Unique Data** | **Commonsense Reasoning** | **Language Understanding** | **Reading Comprehension** | **Symbolic Problem** | **World Knowledge** | **Core**    |
> | -- | -- | -- | -- | -- | -- | -- | -- | -- |
> | Organic | 144B  | 14.4B + 14.4B | 0.41490  | 0.47105 | 0.30908 | 0.19259 | 0.36491 | 0.35390 |
> | WRAP | 144B | 14.4B + 14.4B | 0.41334 | 0.44467  | 0.27998 | 0.19308 | 0.35327 | 0.33999 |
> | ProX | 144B | 14.4B + 14.4B | 0.37050  | 0.47198 | 0.31033 | 0.19441 | 0.37112 | 0.35688 |
> | ReWire | 144B | 14.4B + 14.4B | 0.40759  | 0.47302 | 0.31280 | 0.19138 | 0.37658 | 0.35632 |
> | RePro | 144B | 14.4B + 14.4B | 0.41768 | 0.47535 | 0.32152 | 0.20401 | **0.37801** | 0.36272 |
> | RePro | 144B | 14.4B + 7.2B  | **0.45065**  | **0.48314** | **0.33544** | **0.20831**  | 0.37045 | **0.37050** |
>
> On the other hand, we also scale down our rephraser by using OLMo-2-0425-1B-Instruct as an additional base model. OLMo is a fully open, academically released model family, making it an ideal choice for testing the generalization ability of our method. As shown in the table below, using OLMo-2-0425-1B-Instruct as the base rephraser yields overall performance comparable to Qwen3-4B in both the 400M and 1B settings and shows a clear advantage on reading comprehension tasks. This result not only demonstrates the robustness of our method but also highlights that even a smaller 1B model can be turned into an effective rephraser through our RL, which helps mitigate concerns about relying on a larger rephraser. The OLMo experiment on the 3B setup is still ongoing, and we will update its results once the run completes.
>
> | **400M Setting** | **Commonsense Reasoning** | **Language Understanding** | **Reading Comprehension** | **Symbolic Problem** | **World Knowledge** | **Core**    |
> |  -- |  -- |  -- |  -- |  -- |  -- | -- |
> | Organic | 0.23613 | 0.27079 | 0.03724 | 0.14535  | 0.20126 | 0.18990 |
> | RePro (Qwen3-4B) | **0.28454** | **0.27792** | 0.07181  | **0.19409** | 0.21154 | 0.21658 |
> | RePro (OLMo2-1B) | 0.27361 | 0.27692 | **0.10292**  | 0.17819  | **0.22057** | **0.21749** |
>
> | **1B Setting**   | **Commonsense Reasoning** | **Language Understanding** | **Reading Comprehension** | **Symbolic Problem** | **World Knowledge** | **Core**    |
> |  -- |  -- |  -- |  -- |  -- |  -- | -- |
> | Organic  | 0.32348  | 0.38371   | 0.19584 | 0.19795  | 0.28745 | 0.28578 |
> | RePro (Qwen3-4B) | **0.36776**   | **0.38519**    | 0.20832  | **0.20597** | **0.30304** | **0.29929** |
> | RePro (OLMo2-1B) | 0.36480 | 0.38379   | **0.23370** | 0.19162 | 0.29755 | 0.29746 |
>
> [1] Kang, Feiyang., et al. "Demystifying Synthetic Data in LLM Pre-training: A Systematic Study of Scaling Laws, Benefits, and Pitfalls." EMNLP 2025.
>
>
>
> > **Suggestion:** The comparison of pretraining data with fossil fuel
>
> **Response:** Thank you for pointing this out. We agree that the fossil fuel analogy is not ideal, since high-quality pretraining data does not disappear after use. Our intention was only to emphasize that the amount of *new* high-quality human-written text available on the public web is limited and not growing at the pace needed for modern pretraining. To avoid this confusion, we will revise the wording and replace the analogy with a more accurate one. For example, we change our first sentence to “High-quality data is a cornerstone of large language model (LLM) pretraining, yet its growth has not kept pace with the needs of frontier models.”

---

### Official Review · Reviewer_H1ux · 2025-11-02

**Soundness:** 4
**Presentation:** 3
**Contribution:** 3
**Rating:** 8
**Confidence:** 4

**Summary:**

The paper introduces a data recycling framework that uses a smaller reinforcement-trained model to automatically rewrite web text into higher-quality pretraining data. Rather than filtering or prompting large models to clean data, the method trains a dedicated “rephraser” that learns to improve readability and consistency while preserving the original meaning. This approach aims to make pretraining corpora both cleaner and more diverse without heavy human curation or large-model prompting. Experiments show that models trained on the rewritten data perform better on downstream benchmarks and that the rephraser operates far more efficiently than prompt-based alternatives.

**Strengths:**

1. Addresses a practical problem in pretraining: improving web data quality without costly human filtering or large-model prompting, using a small reinforcement-trained rephraser.
2. Computational savings over prior works that use large models to rephrase the text, while recovering most of the performance that those methods gave
3. Ablations confirm that each reward component (quality and faithfulness) is necessary to prevent drift and reward hacking, supporting the method’s design.

**Weaknesses:**

1. It would be nice to see a clearer / more varied perspective on what the rephrased text looks like. Concerns involve degenerate / repeated formatting, diversity collapse, etc.
2. The benchmark gains are minimal in some cases (requiring 3 decimal places to see the value of it). I would love to see some discussion of what to take away from the behavior on  "core" tasks vs the other ones.
3. In some sense, the place where we need more / better data is in specialized domains like math and code. I think the authors can emphasize that their method enables straightforward adaptation of any synthetic data generator to maximize some reward (eg in math, maybe it is "diverse CoTs")

**Questions:**

See above

---

> ### Author Response · Authors · 2025-11-22
> **Official Rebuttal to Reviewer H1ux**
>
> Thank you for your time and insightful review of our paper! We address your questions/comments below:
>
> > **Weakness 1:** A more varied perspective on what the rephrased text looks like. Concerns involve degenerate / repeated formatting, diversity collapse
>
> **Response:** Thank you for your suggestion. To better understand the rephrasing behavior of RePro, We provide more representative examples in Appendix D.2, where RePro transforms organic data into higher-quality rephrasings.
>
> - For low-quality organic data, we observe that RePro can clarify unclear information or logical gaps.
>
> - For moderate-to-low quality data that often contains irrelevant webpage artifacts (such as navigation text or image placeholders), RePro removes these elements during rephrasing while keeping the essence of the text.
>
> - For already high-quality data, RePro reorganizes and paraphrases the content to improve clarity and further raise the quality score.
>
> Across these cases, as well as many more examined through manual inspection, we did not observe issues such as degeneration, repeated formatting patterns, or reduced diversity. Through RL optimization, RePro preserves the key characteristics of organic data (writing style, structure, genre, etc.) and learns to flexibly apply various operations to polish the text.
>
> > **Weakness 2:** Benchmark gains are minimal in some cases. Some discussion of what to take away from the behavior on "core" tasks vs the other ones
>
> **Response:** Thank you for your suggestion. First, we would like to note that DCLM is a standardized and rigorous pretraining data curation benchmark, with 22 Core tasks that holistically evaluate pretrained models, so improvements on it reflect meaningful and robust gains, rather than being limited to isolated tasks. Second, our method RePro **doubles** the improvement achieved by the state-of-the-art web recycling baseline (ReWire) over the organic-only baseline in all setups. In addition, RePro has a substantial efficiency advantage (36.7× inference speedup) than ReWire, since we use a much smaller rephraser (4B) compared to ReWire’s 70B model. Third, our performance with a **72B** organic data pool surpasses the organic-only baseline with a **4×72B=288B** data pool (Figure 1), demonstrating a substantially better utilization of existing organic data that directly helps alleviate the “data wall” challenge in LLM pretraining. Together, these results validate that our gains are both significant and consistent.
>
> A more detailed per-category discussion is as follows.
>
> - For commonsense reasoning and symbolic problem solving tasks, a coherent and consistent logical flow within the rephrased text is preferred. This is one of RePro’s strengths, as illustrated in Figure 8 (DataMan criteria). ProX also improves these tasks by removing irrelevant content, but RePro performs better by flexibly applying a wider range of operations, as shown in Section 5.5.
>
> - For language understanding tasks, RePro is notably the only web recycling method that surpasses the organic-only baseline. We hypothesize that this is because RePro is the most faithful method, so the rephrased data resembles organic human-written text in terms of content and writing style. This aligns naturally with the distribution of language understanding tasks, which evaluate a model’s ability to interpret human text. In contrast, other web recycling methods do not explicitly optimize for faithfulness, leading their distributions to drift toward LM-generated text.
>
> - For reading comprehension tasks, rephrasings with explicit contextual explanation may help. For instance, ReWire (which performs well on reading comprehension) often adds phrases such as “To understand the context of the...”, which can provide additional hints, as shown in Appendix D.2. However, repeating similar patterns may reduce the diversity of the training data and limit generalization.
>
> - For world knowledge tasks, RePro outperforms all other baselines but is slightly behind ReWire. This is unsurprising because ReWire uses a much larger and stronger model (Llama-3.3-70B-Instruct) for rephrasing, and prompting such a powerful model can introduce additional factual knowledge from the LLM distillation. However, this also risks reduced faithfulness, which is a known issue in prior studies. RePro condenses the knowledge by removing irrelevant content and adding necessary clarifications, as illustrated in our cases.
>
> We will add these in the paper as well.

---

> > ### Author Response · Authors · 2025-11-22
> > **Official Rebuttal to Reviewer H1ux (Continued)**
> >
> > >  **Weakness 3:** Straightforward adaptation of any synthetic data generator to maximize domain-specific rewards
> >
> > **Response:** Thank you for your suggestion. For domain specific data synthesis, we can indeed apply a similar reinforcement learning framework to train a synthetic data generator that produces more high-quality data. The key difference is that the generator would not be limited to rephrasing, but would create new examples from some task specific seeds, which may vary the reward design. In the math setting, for instance, a correctness reward can be obtained by verifying the final answer; a reasoning quality reward can be provided by an LLM-as-a-judge that scores whether each step is logically sound and whether the reasoning flows coherently; a diversity reward can be computed by detecting different cognitive behaviors [1] in the chain of thought and giving a bonus when the generator produces a reasoning trace that uses a behavior that is currently underrepresented in the current dataset.
> >
> > Our work focuses on recycling web data to help alleviate the “data wall” in general purpose pretraining, but the novel idea of using RL to train the synthetic data generator can indeed extend to specialized domains by choosing suitable reward signals. We will add this discussion in the paper and hope our work can inspire future efforts in this direction.
> >
> > [1] Gandhi, Kanishk, et al. "Cognitive Behaviors that Enable Self-Improving Reasoners, or, Four Habits of Highly Effective STaRs." COLM 2025.

---

### Official Review · Reviewer_XtdU · 2025-11-02

**Soundness:** 3
**Presentation:** 3
**Contribution:** 3
**Rating:** 6
**Confidence:** 4

**Summary:**

The authors study the rephrasing of pretraining data to increase the availability of high-quality data. They design one quality reward and three faithfulness rewards to guide the training of LM rephraser. They demonstrate their method using a 4B rephraser can outperform state-of-the-art methods using 70B models.

**Strengths:**

1. They demonstrated the viability of using small models to obtain high quality recycled data, while this is also shown by some concurrent works.

**Weaknesses:**

1. The novelty and insight feel somehow limited. One can follow up with even more design of rewards to potentially further improve the performance, while how much this could help the community is debatable
2. The scale of the pretrained model is rather limited. Notably, much smaller than the rephraser, which might impact the transfer of knowledge,

**Questions:**

1. Could you show some results that the baselines perform badly with the 4B models? (Even though this may be quite imaginable) Additionally, could you show some result using other model families as the rephraser, preferably the one used by the baseline (llama)?
2. Do you have some preliminary results with slightly larger pretrained models? In the literature of knowledge distillation, which is different but still somewhat related to your research direction, some has shown that distilling from a much larger teacher model might be worse than a smaller one due to gap in model capacity [1]. In your case, similarly, the small pretrained model might not be able to learn/differentiate the content generated from a larger and potentially better rephraser.

[1] Mirzadeh, S. I., Farajtabar, M., Li, A., Levine, N., Matsukawa, A., & Ghasemzadeh, H. (2020, April). Improved knowledge distillation via teacher assistant. In Proceedings of the AAAI conference on artificial intelligence (Vol. 34, No. 04, pp. 5191-5198).

---

> ### Author Response · Authors · 2025-11-22
> **Official Rebuttal to Reviewer XtdU**
>
> Thank you for your time and insightful review of our paper! We address your questions/comments below:
>
> > **Weakness 1:** Novelty and insight of RePro
>
> **Response:** Our work aims to address a critical problem in LLM pretraining: the limited supply of high-quality organic data. This motivates the urgent need for an effective and efficient method to recycle the existing organic pool. Regarding novelty, we introduce a carefully designed set of reward functions for training a faithful rephraser tailored for web recycling, and we systematically examine how different rephrasing methods affect the faithfulness of the recycled data. Our results validate that a relatively small model can be turned into an efficient and high-performing rephraser through RL, and our analyses highlight that preserving the key characteristics of organic data is crucial both for maintaining distributional alignment and for improving pretraining performance.
>
> Building on top of our framework, it is straightforward to explore more customized reward designs for effective utilization of organic data. Beyond this, our RL-based synthetic data generation could be applied not only during pretraining, but also in mid-training or post-training stages where more domain-specific data is needed and a dedicated synthetic data generator could have great potential. Our findings also point toward the possibility of self-improving models, where even smaller pretrained models can be trained to generate data for themselves, offering a new path to mitigate long-term data scarcity. These directions are beyond the scope of this paper, but we have added a discussion of them and believe they represent valuable follow-up opportunities for the community.
>
> > **Question 1-1:** Baseline methods with Qwen3-4B
>
> **Response:** Thank you for your suggestion. We adopt the prompts of the baseline methods (WRAP and ReWire) and use Qwen3-4B as the rephraser. ProX is an operation-based refiner that requires additional supervised training from a 70B LLM, and these supervision signals are not released, so replicating ProX with Qwen3-4B is not feasible now. As shown in the table below, WRAP with Qwen3-4B achieves about a 1% relative gain compared to using Mistral-7B-Instruct-v0.1, and ReWire with Qwen3-4B underperforms Llama-3.3-70B-Instruct. In general, these observations are reasonable, as a stronger model can perform the rephrasing task better due to its stronger instruction-following ability, but prompting alone still remains difficult to control, so the resulting gaps are always limited. From another perspective, modifying the prompt for the same rephraser (Qwen3-4B) also leads to marginal performance changes. Taken together, these results highlight the importance of RL, which provides a more effective and reliable way to optimize the rephraser for web recycling.
>
> | **400M Setting** | **Rephraser**            | **Commonsense Reasoning** | **Language Understanding** | **Reading Comprehension** | **Symbolic Problem** | **World Knowledge** | **Core**    |
> | ---------------- | ------------------------ | ------------------------- | -------------------------- | ------------------------- | -------------------- | ------------------- | ----------- |
> | WRAP             | Mistral-7B-Instruct-v0.1 | 0.24784                   | 0.25798                    | 0.06269                   | 0.16303              | 0.20067             | 0.19536     |
> | WRAP             | Qwen3-4B                 | 0.25928                   | 0.26897                    | 0.02412                   | 0.17616              | 0.20051             | 0.19761     |
> | ReWire           | Llama-3.3-70B-Instruct   | 0.24051                   | 0.26453                    | 0.06232                   | 0.17392              | **0.21246**             | 0.20125     |
> | ReWire           | Qwen3-4B                 | 0.24105                   | 0.26162                    | 0.05101                   | 0.17632              | 0.20887             | 0.19872     |
> | RePro            | Tuned Qwen3-4B                 | **0.28454**               | **0.27792**                | **0.07181**               | **0.19409**          | 0.21154         | **0.21658** |

---

> ### Author Response · Authors · 2025-11-22
> **Official Rebuttal to Reviewer XtdU (Continued)**
>
> > **Question 1-2:** Use other model families as the rephraser
>
> **Response:** Thank you for your suggestion. To generalize our methods, we take OLMo-2-0425-1B-Instruct as an additional base model for our rephraser. OLMo is a fully open, academically released model family, making it an ideal choice for testing the generalization ability of our method (in contrast, Llama is still industry released, and its training data distribution is not fully disclosed). As shown in the table below, using OLMo-2-0425-1B-Instruct as the base rephraser yields overall performance comparable to Qwen3-4B in both the 400M and 1B settings and shows a clear advantage on reading comprehension tasks. This result not only demonstrates the robustness of our method but also highlights that even a smaller 1B model can be turned into an effective rephraser through our RL, which further reduces the cost of web recycling.
>
> | **400M Setting** | **Commonsense Reasoning** | **Language Understanding** | **Reading Comprehension** | **Symbolic Problem** | **World Knowledge** | **Core**    |
> | ---------------- | ------------------------- | -------------------------- | ------------------------- | -------------------- | ------------------- | ----------- |
> | Organic | 0.23613 | 0.27079 | 0.03724 | 0.14535              | 0.20126             | 0.18990     |
> | RePro (Qwen3-4B) | **0.28454** | **0.27792** | 0.07181  | **0.19409** | 0.21154 | 0.21658     |
> | RePro (OLMo2-1B) | 0.27361 | 0.27692 | **0.10292**  | 0.17819              | **0.22057** | **0.21749** |
>
> | **1B Setting**   | **Commonsense Reasoning** | **Language Understanding** | **Reading Comprehension** | **Symbolic Problem** | **World Knowledge** | **Core**    |
> | ---------------- | ------------------------- | -------------------------- | ------------------------- | -------------------- | ------------------- | ----------- |
> | Organic          | 0.32348                   | 0.38371                    | 0.19584 | 0.19795              | 0.28745             | 0.28578     |
> | RePro (Qwen3-4B) | **0.36776**               | **0.38519**                | 0.20832  | **0.20597** | **0.30304** | **0.29929** |
> | RePro (OLMo2-1B) | 0.36480 | 0.38379                    | **0.23370** | 0.19162 | 0.29755             | 0.29746     |
>
> >  **Weakness 2 & Question 2:** Results with slightly larger pretrained models
>
> **Response:** Thank you for your suggestion. To scale up our findings, we follow the DCLM 3B-1x setup (2.8B model, 55.9B tokens) and report the results below (also in the updated PDF). RePro still holds the leading position among all web recycling methods, doubling the gains of the second-best baseline (ProX). We also observe that in this larger setup, Wikipedia-style rephrasing (WRAP) can even harm performance, likely due to its limited diversity and weaker preservation of the original distribution, both of which become increasingly important for larger pretraining models.
>
> We would like to note that, compared to the 400M setup, the 3B setup is **undertrained** due to our compute constraints, which may limit the absolute performance gains across all methods. Therefore, we also follow the optimal organic-to-rephrased mixture ratio (2:1) identified by prior work [1] to better reveal the potential of our method. Under this configuration, RePro achieves a **4.7%** relative gain over the organic-only baseline. In summary, RePro consistently outperforms existing baselines in a larger pretraining setup and shows strong scalability and generalization ability of a relatively small but well-optimized rephraser.
>
> | **3B Setting** | **Pool** | **Unique Data** | **Commonsense Reasoning** | **Language Understanding** | **Reading Comprehension** | **Symbolic Problem** | **World Knowledge** | **Core**    |
> | -------------- | -------- | --------------- | ------------------------- | -------------------------- | ------------------------- | -------------------- | ------------------- | ----------- |
> | Organic | 144B     | 14.4B + 14.4B   | 0.41490  | 0.47105 | 0.30908 | 0.19259 | 0.36491 | 0.35390     |
> | WRAP | 144B     | 14.4B + 14.4B   | 0.41334 | 0.44467                    | 0.27998 | 0.19308 | 0.35327 | 0.33999     |
> | ProX | 144B     | 14.4B + 14.4B   | 0.37050                   | 0.47198 | 0.31033 | 0.19441 | 0.37112             | 0.35688     |
> | ReWire         | 144B     | 14.4B + 14.4B   | 0.40759                   | 0.47302 | 0.31280 | 0.19138              | 0.37658 | 0.35632     |
> | RePro | 144B     | 14.4B + 14.4B   | 0.41768 | 0.47535 | 0.32152 | 0.20401              | **0.37801**         | 0.36272     |
> | RePro | 144B     | 14.4B + 7.2B    | **0.45065**               | **0.48314** | **0.33544** | **0.20831**          | 0.37045 | **0.37050** |
>
> [1] Kang, Feiyang, et al. "Demystifying Synthetic Data in LLM Pre-training: A Systematic Study of Scaling Laws, Benefits, and Pitfalls." EMNLP 2025.

---

### Author Response · Authors · 2025-12-03
**Author Final Remarks**

Dear ACs and reviewers,

We sincerely thank all four reviewers for their thoughtful and constructive feedback. Reviewers highlighted multiple strengths of our work, including the practical motivation of improving web-data quality (`H1ux`), a clear RL formulation with well-designed reward signals to train a faithful rephraser (`H1ux`, `A8Ei`, `J499`), consistent improvements over prompting-based and operation-based baselines with smaller costs (`XtdU`, `H1ux`, `A8Ei`, `J499`), and comprehensive ablations supporting each component (`H1ux`, `A8Ei`, `J499`). Taken together, we appreciate the recognition that RePro provides a principled and scalable approach for the faithful recycling of organic data, a practical direction for addressing emerging data-quality limitations in LLM pretraining.

Across reviews, a shared concern was the generalization to different rephraser families and scalability to larger pretraining setups (`XtdU`, `A8Ei`). To address this concern, we incorporated **OLMo-2-1B** as a smaller rephraser and **DCLM 3B-1x** (Table 2) as a larger pretraining setup. Using OLMo-1B in RePro achieves **comparable** gains to Qwen3-4B, demonstrating the strong robustness of our method using different rephrasers. Meanwhile, RePro achieves a **4.7%** relative improvement over the organic-only baseline in the DCLM 3B-1x setup, thus **doubling** the state-of-the-art method’s gains across all three setups. These results confirm that RePro generalizes effectively across both rephraser families and pretraining model scales.

Another recurring point was whether the gains originate from RL rather than stronger prompting (`A8Ei`, `J499`). In our rebuttal, we added **enhanced prompting and SFT baselines** with explicit reward descriptions (Table 3), as well as **sampling ablations** with temperature and top-p variations. These strengthened baselines achieve only marginal improvements and remain **clearly below RePro**, confirming that **RL is much more effective** for refining the rephraser than prompting alone.

Reviewer `A8Ei` also asked whether RePro truly recycles low-quality organic data. In response, we incorporated new analyses on recycling **non-high-quality** data in Section 5.3. We observed that **63.5%** of the high-quality RePro recycled data comes from low-quality organic sources, while the best baseline number is **42.5%**. Furthermore, training only on rephrasings of low-quality organic data recovers **~80%** of RePro’s full gains. Our updated Figure 3 further validates that RePro produces more high-quality tokens than baselines from the same organic pool. These results demonstrate that RePro substantially upgrades low-quality text, making them useful in the pretraining. We also included representative examples in Appendix D.1 illustrating how RePro transforms various organic data into higher-quality rephrasings.

In summary, our rebuttal addressed concerns regarding scalability, RL necessity, and low-quality data recycling, while reinforcing the strengths highlighted by reviewers. We believe RePro enables a novel and practical path toward mitigating the data-quality bottleneck in LLM pretraining.

For your convenience, we summarize the changes in our revised paper:

1. **DCLM 3B-1x results** (Table 2)
2. **Enhanced prompting and SFT baselines** (Table 3)
3. **Baseline comparisons across unique token budgets** (Figure 3)
4. **Recycling of non-high-quality data** (Section 5.3)
5. **Representative rephrasing examples** (Appendix D.1)

We will incorporate all other experiments used in the rebuttal into the next version. Thank you again for your valuable time and thoughtful consideration!

---

### Meta-Review · Area_Chair_LDzj · 2026-01-09

**Summary:**

The following major concerns have been adequately addressed:
-  generalization to different rephraser families and scalability to larger pretraining setups.
- the gains originate from RL rather than stronger prompting.
- Is RePro truly recycling low-quality organic data?

The following concerns have not been addressed
- The contribution is similar to existing approaches
- Performance Gains vs. Methodological Complexity trade-off doesn't seem favorable

**Reviewer Concerns:**

The following major concerns have been adequately addressed:
-  generalization to different rephraser families and scalability to larger pretraining setups.
- the gains originate from RL rather than stronger prompting.
- Is RePro truly recycling low-quality organic data?

**Reviewer Scores:**

I believe
- Reviewer XtdU would keep their score
- Reviewer H1ux would keep their score
- Reviewer A8Ei would raise their score
- Reviewer J499 would keep their score

---

### Decision · Program_Chairs · 2026-01-26

Reject